



**Measurement report: Enhanced photochemical formation of**
**formic and isocyanic acids in urban region aloft: insights**
**from tower-based online gradient measurements**
Qing Yang[1,2], Xiao-Bing Li[1,2,*], Bin Yuan[1,2,*], Xiaoxiao Zhang[1,2], Yibo Huangfu[1,2], Lei
Yang[1,2], Xianjun He[1,2], Jipeng Qi[1,2], Min Shao[1,2]
[1] Institute for Environmental and Climate Research, Jinan University, Guangzhou
511443, China
[2] Guangdong-Hongkong-Macau Joint Laboratory of Collaborative Innovation for
Environmental Quality, Guangzhou 511443, China
[*] Corresponding authors: Xiao-Bing Li (lixiaobing@jnu.edu.cn), Bin Yuan
(byuan@jnu.edu.cn)





**Abstract**
Formic acid is the most abundant organic acid in the troposphere and has
significant environmental and climatic impacts. Isocyanic acid poses severe threats to
human health and could be formed through the degradation of formic acid. However,
the lack of vertical observation information has strongly limited the understanding of
their sources, particularly in urban regions with complex pollutant emissions. To
address this issue, continuous (27 days) vertical gradient measurements (five heights
between 5-320 m) of formic and isocyanic acids were made based on a tall tower in
Beijing, China in summer of 2021. Results show that the respective mean mixing ratios
of formic and isocyanic acids were 1.3±1.3 ppbv and 0.28±0.16 ppbv at 5 m and were
2.1±1.9 ppbv and 0.43±0.21 ppbv at 320 m during the campaign. The mixing ratios of
formic and isocyanic acids were substantially enhanced in daytime and correlated with
the diurnal change of ozone. Upon sunrise, the mixing ratios of formic and isocyanic
acids at different heights simultaneously increased even in the residual layer. In addition,
positive vertical gradients were observed for formic and isocyanic acids throughout the
day. The afternoon peaks and positive vertical gradients of formic and isocyanic acids
in nighttime indicate their dominant contributions from photochemical formations.
Furthermore, the positive vertical gradients of formic and isocyanic acids in daytime
imply the enhancement of their secondary formation in urban regions aloft,
predominantly due to the enhancements of oxygenated volatile organic compounds.
The formation pathway of isocyanic acid through HCOOH-CH$_3$NO-HNCO was
enhanced with height but only accounted for a tiny fraction of its ambient abundance.
The abundance and source contributions of formic and isocyanic acids in the
atmospheric boundary layer may be highly underestimated when being derived from
their ground-level measurements. With the aid of numerical modeling techniques,
future studies could further identify key precursors that drive the rapid formation of
formic and isocyanic acids, and quantitatively assess the impacts of the enhanced
formation of the two acids aloft on their budgets at ground level.





## 1. Introduction

Formic acid (HCOOH) is the simplest but the most abundant organic acid in the
troposphere. It has been widely measured in aqueous (clouds and aerosols) and gaseous
phases over urban, rural, and remote regions *(Kawamura and Kaplan, 1983; Chebbi*
*and Carlier, 1996; Kesselmeier et al., 1998; Yu, 2000)*. As important contributors to the
acidity of precipitation, formic and acetic acids can account for 60% of the free acidity
in remote regions *(Galloway et al., 1982; Andreae et al., 1988)*, and over 30% of the
free acidity in heavily polluted regions *(Keene and Galloway, 1984)*. Formic acid is
also an important sink of hydroxyl radicals (OH) in clouds *(Jacob, 1986)*, playing vital
roles in modulating the atmospheric aqueous-phase chemistry through changing pH-
dependent reaction rates of related constituents. An in-depth understanding of the
concentration levels, spatiotemporal variations, and sources of formic acid is key to
elucidating the formation mechanisms of atmospheric secondary pollution. However,
the sources and sinks of atmospheric formic acid are still poorly understood so far.
There have been many reported sources of atmospheric formic acid. Primary
emissions from vegetation activity *(Andreae et al., 1988; Kesselmeier et al., 1998)*,
microbial metabolism *(Enders et al., 1992)*, biomass burning *(Goode et al., 2000)*, and
vehicle exhaust *(Kawamura et al., 2000)* were identified as important sources of formic
acid. Secondary formation from photochemical degradation of volatile organic
compounds (VOCs) is another significant source of formic acid *(Khare et al., 1999;*
*Veres et al., 2011; Le Breton et al., 2014; Liggio et al., 2017)*. However, current
chemical transport models still highly underestimate ambient concentrations of formic
acid *(Stavrakou et al., 2011; Paulot et al., 2011; Millet et al., 2015)* and cannot well
reproduce its vertical variations. For example, Mattila et al. *(2018)* measured vertical
profiles of formic acid using an elevator on the Colorado Front Range BOA tower. They
found that formic acid mixing ratios generally decreased with height throughout the day,
but there were no known sources to explicitly explain the net surface emissions. The
vertical distribution and variation patterns of formic acid in the atmospheric boundary



layer can provide valuable information on the identification and determination of source
contributions. Nevertheless, the vertical variations and key drivers of formic acid,
particularly in urban regions, are still unclear due to the lack of adequate vertical
observations.
Isocyanic acid (HNCO) is an inorganic acid and has attracted extensive concerns
worldwide in recent years due to its strong toxicity *(Wang et al., 2007; Jaisson et al.,*
*2011; Koeth et al., 2013)*. Previous studies have reported that isocyanic acid is highly
soluble at physiological pH and the dissociated cyanate ions (NCO⁻) are closely linked
to atherosclerosis, cataracts, and rheumatoid arthritis *(Mydel et al., 2010; Roberts et al.,*
*2011)*. At present, there is no standard to clearly define the critical levels of isocyanic
acid pollution in ambient air *(Rosanka et al., 2020)*. A mixing ratio of 1 ppbv was
considered the upper limit of ambient isocyanic acid, which is derived from the
threshold of protein carbamylation reactions initiated by HNCO or NCO⁻ in human
body *(Verbrugge et al., 2015; Fulgham et al., 2020)*. Similar to formic acid, our
understanding of isocyanic acid sources is also very limited.
As reported in the literature, primary emissions of isocyanic acid are mainly from
combustion sources including cigarette smoke *(Hems et al., 2019)*, gasoline and diesel
engine exhausts *(Wren et al., 2018)*, and biomass combustion *(Wentzell et al., 2013; Li*
*et al., 2021; Chandra and Sinha, 2016)*. Wet and dry deposition is known as the main
sink of isocyanic acid *(Roberts et al., 2014; Rosanka et al., 2020)*. In addition, isocyanic
acid is highly soluble at atmospheric pH and can be hydrolyzed to $NH_3$ and $CO_2$ *(Zhao*
*et al., 2014; Roberts and Liu, 2019)*. Secondary formation is another important source
of atmospheric isocyanic acid and the known precursors include amides *(Barnes et al.,*
*2010)*, urea *(Jathar et al., 2017)*, and nicotine *(Roberts et al., 2011; Borduas et al.,*
*2016)*.
Amides are reported to be the main precursors of isocyanic acid in urban region
*(Wang et al., 2020)*. Isocyanic acid is the oxidative degradation product of amides
initiated by OH radicals, $NO_3$, radicals, and Cl atoms *(Barnes et al., 2010)*. In addition



to primary emissions from organic solvents and various industrial processes, amides
can be also formed through the atmospheric accretion reactions of organic acids with
amines or ammonia *(Barnes et al., 2010; Yao et al., 2016)*. The degradation of formic
acid may be an important formation pathway of isocyanic acid in the atmosphere. For
example, formamide could be formed from the atmospheric accretion reactions of
formic acid with amines or ammonia and then produce isocyanic acid through reactions
with OH radicals, $NO_3$, radicals, and Cl atoms. The vertical variations of formic acid
will also have vital impacts on the sources and vertical distributions of isocyanic acid
if the above-mentioned speculation is true. Unfortunately, the vertical distributions of
isocyanic acid are also poorly understood due to the lack of measurements.

Chemical ionization mass spectrometer (CIMS) can effectively detect and

quantify atmospheric formic and isocyanic acids *(Bannan et al., 2014; Chandra and
Sinha, 2016; Liggio et al., 2017; Mungall et al., 2018; Fulgham et al., 2019)*. However,
the big mass, large volume, and strict operation environments of CIMS limit its wide
application in making vertical measurements of formic and isocyanic acids. CIMS has
been widely used onboard aircraft or on towers to make online vertical measurements
of formic and isocyanic acids *(Liggio et al., 2017; Mattila et al., 2018)*. Aircraft can
carry many types of instruments and achieve measurements of a large suite of
parameters *(Benish et al., 2020; Zhao et al., 2021)*, but the cost is also very expensive.
Towers can provide vertical observations of target species by setting up sites at different
heights, building mobile platforms (elevators or baskets) *(Mattila et al., 2018)*, and
drawing air from multiple heights to the ground-based instruments through long tubes
*(Hu et al., 2013; Yáñez-Serrano et al., 2018)*. The usage of long tubes is the most
convenient and cost-effective method to make gradient measurements of target gaseous
species so far. However, interactions between gaseous species and tubing walls may
bring unexpected uncertainties for their measurements *(Helmig et al., 2008a; Helmig
et al., 2008b; Schnitzhofer et al., 2009; Sweeney et al., 2010; Pagonis et al., 2017)*.
Therefore, the impacts of long tubing on measurements of formic and isocyanic acids



need to be elucidated.
In this study, we first assessed the effects of long perfluoroalkoxy alkane (PFA)
Teflon tubes on measurements of formic and isocyanic acids. Vertical gradient
measurements of the two acids were made through long tubes on a tall tower in urban
Beijing, China. Then, the vertical variations and sources of the two acids were
investigated and discussed. At last, key conclusions and implications of this study were
summarized.

## 2. Methods and materials


### 2.1. Site description and field campaign


Vertical gradient measurements of gaseous species were made on the Beijing
Meteorological Tower, which is located on the campus of the Institute of Atmospheric
Physics (IAP), Chinese Academy of Sciences. Beijing is the capital city of China with
a population of over 20 million by 2020. Beijing has large anthropogenic emission
intensities and is suffering from severe air pollution problems *(Acton et al., 2020; Meng*
*et al., 2020; Tan et al., 2022)*. The tower is located in the northern part of downtown
Beijing between the 3$^{rd}$ and 4$^{th}$ Ring Roads and is surrounded by urban roads,
expressways, residential areas, restaurants, urban landscaping, and parks. As a result,
concentrations of the primary pollutants at the tower site are mainly contributed by both
anthropogenic (e.g., vehicular exhausts, cooking, and household volatile chemical
products) and biogenic emissions. Detailed descriptions of the tower have been
provided in previous studies *(Acton et al., 2020; Yan et al., 2021)* and will not be
repeated here. The field campaign was carried out from July 17$^{th}$ to August 3$^{rd}$, 2021.

### 2.2. Instrumentation


To obtain online gradient measurements of atmospheric trace gases, we
established a tower-based observation system using a combination of online
measurement techniques and long tubes. The system and related assessments on the



usage of long tubes have been explicitly described in our previous study *(Li et al., 2023)*
and will be briefly introduced here. After removing fine particles by PFA Teflon filters,
ambient air at four altitudes on the tower (namely 47, 102, 200, and 320 m) was
simultaneously and continuously drawn to the ground through long PFA Teflon tubes
(100, 150, 250, and 400 m; outer diameter: 1/2"; inner diameter: 0.374") by a pump.
The flow rate of the sample stream in each tube was controlled by a critical orifice and
ranged between 14-19 standard liters per minute (SLPM). The flow rates in long tubes
were retained as large as possible if instruments allowed to minimize the impact of gas-
surface interactions on measurements of targeted gaseous species *(Deming et al., 2019;*
*Li et al., 2023)*. Two air-conditioned containers were placed next to each other on the
base of the tower and all the instruments were operated inside. An additional inlet of
the tube was mounted on the rooftop of the container (approximately 5 m above ground
level) to make measurements of trace gases near the surface. Therefore, the tower-based
observation system consisted of five inlet heights ranging from the ground level to 320
m. Inlets of the instruments were connected to the outlet of a Teflon solenoid valve
group, which was used to perform the switch of the inlet heights at time intervals of 4
minutes. Vertical gradient measurements of gaseous species were cyclically made over
periods of 20 minutes. Indoor PFA Teflon tubes were wrapped with insulation tubes and
were heated to prevent condensation of water and organic gases.

Formic and isocyanic acids were measured by a high-resolution time-of-flight

chemical ionization mass spectrometer with iodide reagent ion (ToF-CIMS). A Filter
Inlet for Gases and AEROsols (FIGAERO) was used to perform the switch between the
gas and particle measurement modes *(Lopez-Hilfiker et al., 2014)*. The ionic molecular
reaction (IMR) chamber is adjacent to the FIGAERO and utilizes a vacuum ultraviolet
ion source (VUV-IS). Iodide anion ($I^-$) is produced from the photoionization of methyl
iodide ($CH_3I$) in MIR *(Ji et al., 2020)*. During the measurements, $I^-$ was produced by
introducing the $CH_3I$ gas standard (1000 ppm, Dalian Special Gases, China) to the IMR
chamber at a flow rate of 2 standard cubic centimeters per minute (SCCM) in 200



SCCM high-purity nitrogen ($N_2$, 99.9995%) by the VUV-IS. The pressure of the IMR
chamber was maintained at 70-80 mbar. Due to the high sensitivity to oxygenated
volatile organic compounds (OVOCs), the iodine ion source has been widely used in
previous studies *(Yuan et al., 2015; Schobesberger et al., 2016; Mungall et al., 2018)*.
Flow rates of the sample gas were maintained at 2 SLPM using a critical orifice. During
the field campaign, the FIGAERO was set to 24 minutes for gas measurements and 36
minutes for particle measurements in one-hour cycles. In gas mode, the first 21 minutes
were used to measure ambient air and the last 3 minutes were used for instrument
background measurements by introducing zero air at 3 SLPM. In addition, instrument
background measurements were also made for 10 s at time intervals of 210 s *(Palm et*
*al., 2019)*.

Calibrations of the ToF-CIMS for formic and isocyanic acids were performed in

the laboratory before and after the field campaign. Standard solutions of formic acid
were evaporated using the liquid calibration unit (LCU, IONICON Analytik GmbH)
and then diluted to designated concentration gradients by being mixed with zero air at
five flow rates. The gas standard of isocyanic acid is unstable at ambient temperature
and thus no commercial gas cylinder was available. Instead, cyanuric acid solution was
put into a diffusion cell and heated to 300 ˚C to generate isocyanic acid gas at a stable
mixing ratio. An ion chromatograph was used to quantify the concentration of the gas
standard by measuring deionized water that absorbed the isocyanic acid gas. Detailed
information about the isocyanic acid calibration procedure has been provided in our
previous work *(Wang et al., 2020)*. Impacts of the changes in ambient humidity on
measurements of the ToF-CIMS for both formic and isocyanic acids were determined
in the laboratory and were eliminated when calculating their respective concentrations.
Measured signals of the ToF-CIMS were processed using the Tofware software package
(version 3.0.3; Tofwerk AG, Switzerland).

A high time-resolution proton-transfer-reaction quadrupole interface time-of-

flight mass spectrometer (PTR-ToF-MS) with both $H_3O^+$ and $NO^+$ ion chemistry was



used to make some precursor measurements of the two acids, such as isoprene,
aromatics, OVOCs, and amides. Detailed information about the configuration and
operation setup of the PTR-ToF-MS has been provided in our previous studies *(Yuan et*
*al., 2017; Wu et al., 2020; Li et al., 2022)*. Mixing ratios of $O_3$, CO, and $NO_2$ were
measured by a UV absorption $O_3$ analyzer (T400, Teledyne API, USA), a gas filter
correlation CO analyzer (T300, Teledyne API, USA), and a trace level NOx analyzer
(42i, Thermos, USA), respectively. Photolysis rates were measured by a PFS-100
photolysis spectrometer (Focused Photonics Inc.) on the rooftop of the container. The
planetary boundary layer height (PBLH) data was obtained from the website of the Air
Resources Laboratory (https://ready.arl.noaa.gov/READYamet.php). Measurements of
isocyanic acid and amides made in Guangzhou and Gucheng in China were also used
in this study for comparison, and more information about these observations can be
found in our previous papers *(Wang et al., 2020)*.

**2.3.  Tubing assessment**

The tower-based observation system used long PFA Teflon tubes (hundreds of
meters in length) to draw air samples from different heights. The interactions between
tubing inner walls and organic compounds, namely the absorption/desorption of trace
gases, have nonnegligible impacts on their measurements after traversing such long
tubes *(Pagonis et al., 2017; Deming et al., 2019)*. The equilibrium between the
absorption and desorption of organic compounds on tubing walls required distinct times,
namely tubing delay, for different species. For nonpolar/weak-polar organic compounds,
their tubing delays and measurement uncertainties after traversing long tubes are
dependent on their saturation concentrations and the flow rates of sample streams but
are independent of changes in humidity *(Krechmer et al., 2017; Pagonis et al., 2017)*.
For some small polar organic compounds, their tubing delays and measurement
uncertainties after traversing long tubes are dependent on Henry's law coefficients and
are affected by changes in humidity *(Liu et al., 2019)*. The performance of long PFA
Teflon tubes in measuring concentrations of nonpolar/weak-polar organic compounds



and inorganic species (e.g., ozone, NO, $NO_2$, and $CO_2$) has been assessed in our
previous work *(Li et al., 2023)*. The impacts of long PFA Teflon tubes on measurements
of formic and isocyanic acids are still unclear and will be assessed in this study.

Long PFA Teflon tubes with an outer diameter of 1/2" and an inner diameter of

0.374" were used to draw air samples from different altitudes and thus were assessed.
At flow rates below 20 SLPM, suitable pressure drops can be maintained in these long
tubes for instrument operation *(Li et al., 2023)*. The effects of long tubes on
measurements of formic and isocyanic acids were mainly assessed using the same
methods in the literature *(Li et al., 2023)*. The tubing delay of formic acid is estimated
as the time required to reach 90% of the concentration change made at the tubing inlet.
The depassivation curve of formic acid measured at the air outlet end of the tubing was
used to calculate its tubing delay and was obtained by using a step-function change in
its concentration at the tubing inlet *(Pagonis et al., 2017; Deming et al., 2019)*. The
formic acid signals were normalized to those measured at the beginning of the step-
function change and then were fitted using the double exponential method, as shown in
Figure 1. Finally, the tubing delay of formic acid was determined when the fitting line
decreased to 0.1. The previous study *(Li et al., 2023)* has reported that inorganic species
have small tubing delays even in a 400 m long tube. Therefore, tubing delays of
isocyanic acid in long tubes are not discussed in this study.

To further assess the impacts of long tubes (namely 100, 200, 300, and 400 m)

on measurements of formic and isocyanic acids in real environments, their ambient
mixing ratios measured through different lengths of tubes were intercompared by
running the inlets side by side at ground level. Ambient air samples were sequentially
drawn with and without the tubes through a Teflon solenoid valve group (Figure S1),
which was set to perform the switch at time intervals of 4 minutes. Instrument
backgrounds of the two species were measured for 10 s at time intervals of 1 minute by
passing zero air into the instrument at a flow rate of 3 SLPM. Inter-comparisons of the
formic acid and isocyanic acid measurements made through different lengths of tubes





were mainly performed using linear fittings (y=$k$x+$b$; $k$ is the slope and $b$ is the
intercept).

## 3. Results and Discussions

### 3.1. Interactions between long tubes and the two acids

As shown in Figure 1, signals of formic acid measured by the ToF-CIMS had a
tubing delay of 23 s after traversing the 400 m long tube at the flow rate of 13 SLPM.
In addition to the interactions between tubing walls and formic acid molecules *(Pagonis*
*et al., 2017; Deming et al., 2019)*, molecular diffusion and dispersion (namely Taylor
dispersion) can cause the longitudinal mixing of gas molecules in the tubing and is also
an important factor contributing to the measured delays *(Sweeney et al., 2010)*.
Molecular diffusion and dispersion have strong dependences on molecular diffusion
coefficients and tubing flow rates *(Karion et al., 2010)*. The influential time of Taylor
dispersion on the measurements of formic acid through a 400 m long tube at the flow
rate of 13 SLPM was estimated to be only 2.9 s, which is much smaller than the
measured tubing delay (23 s) of formic acid. Therefore, the adsorption/desorption of
formic acid molecules on tubing inner walls plays a dominant role in determining the
tubing delay.
For most organic compounds, the tubing delays generally depend on tubing flow
rates and their saturated concentrations ($C$*) *(Li et al., 2023; Deming et al., 2019)*. With
the increase in tubing length and flow rate, the tubing delays of organic compounds will
rapidly decrease *(Liu et al., 2019)*. Therefore, the tubing flow rates should be as large
as possible if the instrument could work normally. In addition, the tubing delays of
organic compounds generally increase with the decrease in their $C$* *(Li et al., 2023)*. It
must be acknowledged that tubing delay is inevitable. The analysis time scales of
species concentrations measured through long tubes should be greater than their tubing
delays, especially for those with small $C$*.
As shown in Figure S2(a), ambient mixing ratios of formic acid measured





through the 400 m long tube varied consistently with those measured without the tube
with mean values of 4.14 and 4.09 ppbv, respectively. The mixing ratios of formic acid
measured with the long tube were slightly higher in the daytime and lower at night in
comparison with those measured without the long tube. We also conducted a correlation
analysis between the mixing ratios of formic acid measured with and without long tubes.
As shown in Figure 2, the mixing ratios of formic acid measured with and without the
400 m long tube agreed within 20%, but the slope of the linear fitting ($k$=0.84) is lower
than 1. The differences of formic acid mixing ratios measured with and without the 400
m long tube were predominantly caused by the long-tail memory effect of the tubing
(Figure 1). For example, the mixing ratios of formic acid measured through the 400 m
long tube at night equaled to its ambient mixing ratios plus those released from the
tubing inner wall. The tubing delay of formic acid was determined when its mixing
ratios reached 90% of the change before entering the tubing. However, the long-tail
memory effect of the tubing mainly focused on the rest 10% of the change (Figure 1),
which required a much longer time to stabilize.

Impacts of the tubing memory effects will be accumulated due to the continuous

change in ambient concentrations of formic acid. To further assess the impacts of tubing
memory effects on measurement uncertainties of the two acids, differences between
mixing ratios of the species X (namely formic and isocyanic acids) measured with and
without long tubes at time $t$ (denoted by $\delta[X]_t$) were calculated using Eq. (1):
$$\delta[X]_t = [X_{without}]_t - [X_{with}]_t \tag{1}$$
where $[X_{with}]_t$ and $[X_{without}]_t$ refer to mixing ratios of the species X measured at
time $t$ with and without long tubes, respectively; $\Delta t$ is the change in time relative to
time $t$ and was used to characterize the influential time of the memory effect. In addition,
the changes in mixing ratios of the species X measured using long tubes at time $t$ relative
to its average mixing ratio over the previous time interval of $\Delta t$ (denoted by $\Delta[X]_t$)
was also calculated using Eq. (2):
$$\Delta[X]_t = [X_{with}]_t - \frac{\sum_{t-\Delta t}^{t}[X_{with}]}{\Delta t} \tag{2}$$



A strong correlation between $\delta[X]_t$ and $\Delta[X]_t$ could be captured at a certain $\Delta t$ if
the tubing memory effect make essential contributions to measurement uncertainties of
the species X after traversing long tubes. For the 400 m long tubing, $\delta[X]_t$ and $\Delta[X]_t$
had the strongest correlation ($R^2$=0.89) when $\Delta t$ was approximately 14 h (Figure S3).
As also shown in Figure 2(a), the mixing ratios of formic acid measured with and
without the 400 m long tube agreed well when $\Delta[HCOOH]$ approached to zero. The
decrease and increase in $\Delta[HCOOH]$ will enlarge measurement uncertainties of formic
acid using the long tube. In morning periods, ambient mixing ratios of formic acid
rapidly increased. As a result, the mixing ratios of formic acid measured through the
400 m long tube were slightly lower than its ambient mixing ratios due to the absorption
of formic acid by tubing inner walls. In evening and nighttime periods, an opposite
phenomenon was observed due to the desorption of formic acid from tubing inner walls
(Figure S2). In addition to the 400 m long tube, impacts of the tubes with lengths of
100, 200, and 300 m on measurements of formic acid were also assessed, as shown in
Figures 2(c) and 3(a). The usage of tubes with lengths of 100, 200, and 300 m has
negligible impacts on the measurements of formic acid.

In contrast to formic acid, the usage of long tubes had minor impacts on the

measurements of isocyanic acid. The mixing ratios of isocyanic acid measured with and
without the 400 m long tube varied consistently ($k$=0.86, $R^2$=0.90) with mean values of
0.25 and 0.26 ppbv, respectively (Figure S2). As shown in Figure 2(b), $\Delta[HNCO]$ is
evenly distributed on both sides of the 1:1 line. Therefore, the changes in ambient
concentrations of isocyanic acid do not have significant impacts on the measurements
of isocyanic acid through the long tubes. As also shown in Figure 3(b), $\delta[HNCO]$ and
$\Delta[HNCO]$ of isocyanic acid were independent of the changes in isocyanic acid mixing
ratios. The $R^2$ values of linear fittings were less than 0.21 for the isocyanic acid
measurements made using different lengths of tubes. This is consistent with the results
reported in the literature *(Helmig et al., 2008a; Helmig et al., 2008b; Li et al., 2023)*
that inorganic species with low reactivities can be well measured using long PFA Teflon



tubes. The test results confirm that the measurements of formic and isocyanic acids
made through long tubes can be used to characterize their vertical and temporal
variations.

### 3.2. Vertical variations and sources of formic acid

Time series of formic acid and ozone mixing ratios at 5 and 320 m are shown in
Figure 4. The concentrations of formic acid and ozone exhibited similar diurnal and
inter-diurnal variations at different altitudes during the campaign. Hourly mean mixing
ratios of ozone exhibited strong temporal variations with an average of 43.5±25.3 ppbv
at 5 m and an average of 53.5±25.0 ppbv at 320 m. Hourly mean mixing ratios of formic
acid at 5 m ranged between 0.1-6.6 ppbv with an average of 1.3±1.3 ppbv at 5 m, which
is comparable to those observed in other megacities, such as Shenzhen (1.2 ppbv) in
China *(Zhu et al., 2019)*, London (1.3 ppbv) in UK *(Bannan et al., 2017)*, and Los
Angeles (2.0 ppbv) in USA *(Yuan et al., 2015)*. By contrast, hourly mean mixing ratios
of formic acid at 320 m had an average of 2.1±1.9 ppbv, approximately 1.6 times higher
than that at 5 m. The temporal variability of formic and isocyanic acids were mainly
caused by the diurnal and inter-diurnal changes in meteorological conditions (e.g., solar
radiation and PBLH).
Before July 12[th], the daily maximum hourly mixing ratios of ozone at 5 m all
exceeded 100 ppbv, indicating the enhanced formation of secondary air pollutants
associated with photochemical reactions. The mixing ratios of formic acid measured
before July 12[th] were also prominently larger than those measured after, suggesting
important contributions from photochemical formations. Because of the precipitation,
weak solar radiation (characterized by small $j(NO_2)$ values) and small PBLHs were
observed from July 13[th] to 30[th], largely suppressing the photochemical formation of
secondary air pollutants. After August 1[st], low mixing ratios of ozone and formic acids
were observed along with the occurrence of favorable dilution conditions characterized
by high PBLHs.
As shown in Figure 5, the mixing ratios of formic acid measured at the five





altitudes (namely 5, 47, 102, 200, and 320 m) exhibited similar diurnal patterns. After
sunrise (~6:00 LT), formic acid mixing ratios increased rapidly at each altitude before
reaching the peak between 14:00-16:00 LT and then continuously declined before
sunrise the following day. Similar diurnal variation patterns of formic acid were also
observed at other urban sites *(Veres et al., 2011)*, rural sites *(Hu et al., 2022)*, and remote
sites *(Schobesberger et al., 2016)*. The diurnal variation patterns of formic acid were
highly similar to those of ozone (a typical secondary pollutant) but were different from
those of VOCs from primary emissions. Taking toluene as an example, toluene is a
typical VOC tracer of anthropogenic emission sources in urban regions, such as
industrial processes and vehicular exhausts *(Fang et al., 2016; Skorokhod et al., 2017)*,
and is also an important precursor of ozone *(Yuan et al., 2012)*. The mixing ratios of
toluene exhibited opposite diurnal variation patterns to those of ozone and formic acids
with the minima occurring at around 14:00 LT. The lower mixing ratios of toluene in
daytime than in nighttime were predominantly caused by the enhancement of
atmospheric dilution and chemical removal by OH radicals *(De Gouw et al., 2018)*. The
mixing ratios of formic acid poorly correlated ($R^2$ ranged between 0.16-0.28) with those
of CO (a typical tracer of combustion sources) at the five altitudes but well correlated
($R^2$ ranged between 0.67-0.75) with those of Ox ($O_3+NO_2$, a conserved metric of ozone
by removing NO titration effect), as shown in Figure 6. These results further confirm
that ambient concentrations of formic acid in urban Beijing were dominantly
contributed by secondary sources associated with photochemical reactions rather than
primary emissions.

Another observed evidence for the dominant contribution of formic acid from

secondary formations is its positive vertical gradients in nighttime (defined as the
period of 22:00-5:00 LT), as shown in Figure 7. Large amounts of formic acid will
accumulate near the surface with strong negative vertical gradients if primary emissions
dominate its contributions, as manifested by vertical toluene profiles. At nighttime, the
mixing ratios of ozone also increased with height due to enhanced removal by NO

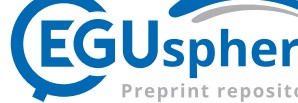



titration and surface dry deposition. The deposition of formic acid was also enhanced
near the surface, driving the formation of positive gradients in vertical formic acid
profiles.
A notable difference existed between the diurnal variation patterns of ozone and
formic acid above the ground. As shown in Figure 5, the mean mixing ratios of ozone
at 5 m rapidly increased from 21.5 ppb to 36.0 ppb from 6:00 to 10:00 LT, while the
mean mixing ratios of ozone at 320 m slightly increased from 16.3 ppbv to 16.9 ppbv
during the same period. As shown in Figure 8, the growth rates of ozone mixing ratios
between 6:00 and 10:00 LT decreased with the increase in height. This phenomenon
indicates relatively weak photochemical ozone formation in urban regions aloft before
10:00 LT due to the lack of reactive ozone precursors (e.g., unsaturated hydrocarbons
and NOx). With the enhancement of the vertical exchange of air masses with the rise of
the boundary layer, large amounts of ozone precursors (e.g., the observed peaks of
toluene mixing ratios at 320 m at 10:00 LT) emitted from surface sources were
transported upward and drove the formation of ozone in high altitudes. In contrast to
ozone, the mixing ratios of formic acid at the five altitudes all increased rapidly between
6:00 and 10:00 LT. The growth rate of formic acid mixing ratios between 6:00 and 10:00
LT kept nearly constant below 320 m (Figure 8). This result implies that the oxidation
products of VOCs over nighttime or in the daytime before are important precursors of
formic acid and can drive the rapid formation of formic acid with further photooxidation.
This speculation can be supported by the vertical and diurnal variations of methyl vinyl
ketone (MVK), methacrolein (MACR), and formaldehyde, which are reported key
precursors of formic acid as shown in Figure 5(d) and 5(e). The diurnal variation
patterns of MVK+MACR and formaldehyde at the five latitudes were nearly the same
with the enhancements in daytime. In addition, concentrations of MVK+MACR and
formaldehyde all increased with height in nighttime and early morning periods,
facilitating the photochemical formation of formic acid even in the residual layer.
As a reactive hydrocarbon species, the mixing ratios of toluene rapidly decreased



with height in daytime (defined as the period of 11:00-16:00 LT, as shown in Figure 7)
due to the combined effects of atmospheric dilution and OH-initiated chemical removal.
By contrast, the mixing ratios of ozone and formic acid increased with height. The
mixing ratios of ozone and formic acid all rapidly increased with height below 102 m,
predominantly attributed to the reduced effect of surface dry deposition with the
increase in height. The mean mixing ratios of formic acid increased by 18% from 102
m to 320 m in daytime, while ozone mixing ratios were well mixed above 102 m. These
observed results support the speculation that photochemical formations of formic acid
were substantially enhanced with the increase in height within the boundary layer.
The precursors and formation mechanisms of atmospheric formic acid have been
extensively investigated in previous studies but still remain uncertain. Isoprene has long
been recognized as an important precursor of formic acid through reactions with $O_3$ and
OH radicals *(Neeb et al., 1997; Paulot et al., 2009)*. Recent studies also found that the
degradation of organic aerosols (OA) derived from isoprene is an important source of
formic acid *(Cope et al., 2021; Bates et al., 2023)*. In addition, the photooxidation of
other biogenic and anthropogenic hydrocarbons is also a key source of formic acid
*(Paulot et al., 2011; Millet et al., 2015)*. Figure 9 illustrates the mean vertical profiles
of several key precursors of formic acid in daytime. The concentrations of isoprene and
toluene (Figure 7) all decreased rapidly with height. By contrast, MVK and MACR, the
primary oxidation products of isoprene *(Grosjean et al., 1993)*, exhibited weak vertical
gradients. Formaldehyde, a more general photooxidation product of VOCs, exhibited
similar vertical distribution patterns to those of ozone. Large amounts of OVOCs were
produced and accumulated in higher altitudes through the oxidation of hydrocarbons
and the further oxidation of some OVOCs during their upward mixing course. MVK,
MACR, and formaldehyde are also key precursors of formic acid. MVK and MACR
can react with $O_3$ to produce formic acid *(Link et al., 2020)*. Formaldehyde can be
converted to methanediol in cloud droplets and then be rapidly oxidized by OH to
produce formic acid *(Franco et al., 2021)*. In addition, enol *(Lei et al., 2020)* and many



other OVOCs (such as glycolaldehyde *(Butkovskaya et al., 2006a)* and hydroxyacetone
*(Butkovskaya et al., 2006b)* can be further oxidized to produce formic acid. Therefore,
high concentrations of OVOCs aloft may be the dominant factor that largely enhances
the photochemical formation of formic acid in urban regions.
As discussed above, formic acid exhibited strong positive vertical gradients
throughout the day, implying that the concentrations of formic acid measured at ground
level were not capable of accurately characterizing its abundance and temporal
variability in the whole boundary layer. Besides, the formic acid formed in daytime and
retained in the nocturnal residual layer also has vital impacts on the budget of formic
acid in the boundary layer. Thus, we used the column-integrated concentration (CIC)
of formic acid (the sum of the abundance in both the nocturnal residual layer and the
boundary layer, see detailed definitions in SI) to further clarify the diurnal variability
in the abundance of formic acid in the boundary layer.
As shown in Figure 4(f), the CICs of formic acid had a flatter diurnal pattern in
comparison to those at ground level. The CICs of formic acid had approximately stable
values overnight and reached a maximum at 16:00 LT. The ratio of the maximum and
minimum of CIC for formic acid was only 1.3, while it was 4.2 for the concentrations
of formic acid at 5 m. These results imply that the removal of atmospheric formic acid
(e.g., surface deposition and various chemical reactions) may be highly overestimated
if only ground-level measurements were used or constrained in numerical models. The
budget of the formic acid in high altitudes in the boundary layer was distinctly different
from those near the surface. As the result, numerical models cannot accurately
reproduce the abundances and budgets of formic acid without the constraints of vertical
observations and the clarification of formic acid formation mechanisms.
**3.3. Vertical variations and sources of isocyanic acid**
The mixing ratios of isocyanic acid also exhibited strong temporal variations
during the campaign with a mean of 0.28±0.16 ppbv at 5 m and a mean of 0.43±0.21
ppbv at 320 m, as shown in Figure 10. The mixing ratios of isocyanic acid measured at



the ground level in urban Beijing were approximately 10 times higher than those
measured in Los Angeles, USA (0.025 ppbv) *(Roberts et al., 2014)* and Calgary, Canada
(0.036 ppbv) *(Woodward-Massey et al., 2014)* but were lower than those measured in
other regions in China. For example, the mean mixing ratio of isocyanic acid was 0.37
ppbv at a rural site (Gucheng) in the North China Plain (NCP), and 0.46 ppbv in urban
Guangzhou in the Pearl River Delta (PRD) region *(Wang et al., 2020)*. Isocyanic acid
will pose a threat to human health when its ambient mixing ratios exceed 1.0 ppbv. In
this study, isocyanic acid mixing ratios greater than 1.0 ppbv were not observed at
ground level but were observed at 320 m on three days. The maximum hourly mixing
ratios of isocyanic acid at 320 m reached 1.63 ppbv at 16:00 LT on July 8[th].
The mixing ratios of isocyanic acid at the five altitudes exhibited similar diurnal
variation patterns. After sunrise, the mixing ratios of isocyanic acid at the five altitudes
all simultaneously increased and peaked at about 14:00 LT. Then, isocyanic acid mixing
ratios decreased slowly and reached the minimum before sunrise the following day.
This diurnal variation pattern of isocyanic acid measured at the ground level in urban
Beijing was not consistent with those measured at the Gucheng site in NCP *(Wang et*
*al., 2020)*. The isocyanic acid mixing ratios at the Gucheng site exhibited insignificant
diurnal variability throughout the day with only a weak morning peak, predominantly
attributed to the enhancement of primary emissions. However, the diurnal variation
patterns of isocyanic acid measured at the five altitudes were well correlated with the
change in solar irradiance and were consistent with those measured at the two sites in
PRD. These results imply that ambient concentrations of isocyanic acid in urban Beijing
were mainly contributed by secondary sources associated with photochemical reactions.
Similar to formic acid, the simultaneous increase of isocyanic acid mixing ratios
at the five altitudes with the onset of sunlight also indicates the presence of adequate
precursors even in the nocturnal residual layer. In addition, the diurnal variability of
isocyanic acid mixing ratios measured below 200 m was much weaker than those
measured at 320 m. For example, the ratio of the daily maximum to the daily minimum



mixing ratios of isocyanic acid was 1.9 at 320 m, while the ratio was only 1.4 at 5 m.
The mean growth rate of isocyanic acid mixing ratios at 320 m (0.05 ppbv h$^{-1}$) between
6:00 and 10:00 LT was approximately five times larger than that at 5 m (0.01 ppbv h$^{-1}$).
The vertical gradients of isocyanic acid between 102 and 320 m were also larger than
those below (Figure 11). The rapid increase in both concentrations and growth rates of
isocyanic acid with height (Figures 8 and 11) implies the enhanced photochemical
formation of isocyanic acid in the middle and upper part of the boundary layer.

Secondary formation precursors of atmospheric isocyanic acid were still poorly

understood so far. Amides were considered important precursors of isocyanic acid
*(Roberts et al., 2014; Rosanka et al., 2020)*. As reported in our previous study *(Wang*
*et al., 2020)*, C$_3$ amides accounted for the largest fraction of the total concentrations of
amides and were dominant contributors to the secondary formation of isocyanic acid.
The mixing ratios of C$_3$ amides in Guangzhou in PRD exhibited strong diurnal
variations. Along with the sunrise, the mixing ratios of C$_3$ amides rapidly decreased and
reached the minimum at 13:00 LT. Afterward, the mixing ratios of C$_3$ amides started to
increase and accumulated at night. However, the mixing ratios of C$_3$ amides in Beijing
and Gucheng in NCP exhibited insignificant diurnal variability, consistent with those
of isocyanic acid. The mean mixing ratios of C$_3$ amides at 5 m in urban Beijing is only
0.03 ppbv during the campaign, which is one order of magnitude lower than those in
Guangzhou (0.35 ppbv) and Gucheng (0.18 ppbv). The mixing ratios of C$_3$ amides
measured at the five altitudes in urban Beijing were also approximately one order of
magnitude lower than those of isocyanic acid (Figure 11). Besides, the mixing ratios of
C$_3$ amides decreased with height in both nighttime and daytime, indicating predominant
contributions from primary emissions. This is consistent with the fact that primary
emissions of chemical composition from industry-related sources have been largely
reduced with the outward migration of industry in urban Beijing. By contrast, the
mixing ratios of isocyanic acid increased with height in both day and night with an
average of 0.32 ppbv at 5 m and 0.60 ppbv at 320 m. These results suggest that C$_3$





amides were far more enough to account for the secondary formation of isocyanic acid
in urban Beijing.
Figure 12(a) gives the composition and average concentrations of $C_1$-$C_{10}$ amides
measured at the five altitudes during the campaign. $C_2$ amides accounted for the largest
fraction of the total mixing ratios of amides. The total mixing ratios of amides exhibited
decreasing tendencies with the increase in height, suggesting predominant contributions
from direct emissions of surface sources. As for formamide, its mixing ratios exhibited
an increasing tendency from 0.024 ppbv at 5 m to 0.030 ppbv at 320 m. The positive
vertical gradients of formamide suggest its enhanced formation with height, probably
due to the enhancements of formic acid. However, the average concentration ratios of
formamide to formic acid slightly varied between 0.01 and 0.02 among the five heights.
The average concentration ratios of formamide to isocyanic acid decreased from 0.09
at 5 m to 0.07 at 320 m. These results imply that the formation of isocyanic acid through
the pathway of HCOOH-CH$_3$NO-HNCO may be enhanced with the increase in height
but could only contribute a tiny fraction of the observed isocyanic acid, as shown in
Figure 12(b). Assuming the full conversion of $C_1$-$C_{10}$ amides to isocyanic acid, the
average concentration ratios of amides (sum of $C_1$-$C_{10}$) to isocyanic acid below 320 m
only ranged between 0.32 and 0.56 and decreased with height. Therefore, in addition to
amides, there must be other important precursors and formation pathways of isocyanic
acid, particularly in high altitudes. The simultaneous increase of isocyanic acid
concentrations at the five heights upon sunrise (Figure 10) implies the presence of
adequate precursors in the nocturnal residual layer. The oxidation products of VOCs
driven by ozone and NO$_3$ radicals in nighttime may be an important class of precursors.
In addition, the largest growth rates and highest concentrations of isocyanic acid at 320
m in daytime also suggest that high concentrations of OVOCs and low-NOx conditions
may enhance the secondary formation of isocyanic acid.
The positive vertical gradients of isocyanic acid imply that the secondary
formation of isocyanic acid aloft could serve as an important source of surface isocyanic



acid in daytime driven by turbulence mixing. The CICs of isocyanic acid were
calculated to further clarify its abundance and temporal variability in the whole
boundary layer. Distinct diurnal patterns were observed between the ground-level
concentrations and CICs of isocyanic acid. Analogous to formic acid, the CICs of
isocyanic acid varied insignificantly over nighttime and enhanced in daytime, reaching
the maximum at approximately 14:00 LT. Therefore, the problems of formic acid
caused by the limitations of ground-level observations also raised for isocyanic acid.

## 576  4. Conclusion

In this study, vertical and diurnal variations of formic and isocyanic acids in

urban Beijing were investigated using tower-based online gradient measurements. The
measurements of isocyanic acid can be well measured through long PFA Teflon tubes.
The measurements of formic acid made through long tubes were slightly influenced by
the memory effect of tubing walls but had minor impacts on analyzing its vertical
distributions. The concentrations of formic and isocyanic acids all increased with height
in both nighttime and daytime. The diurnal and vertical distribution patterns of formic
and isocyanic acids all suggest that their abundances in the boundary layer were
dominantly contributed by secondary formation associated with photochemical
reactions. The photochemical formations of formic and isocyanic acids were also
substantially enhanced with the increase in height. The formation pathway of isocyanic
acid through HCOOH-CH$_3$NO-HNCO only accounted for a tiny fraction of its ambient
abundance. The formic and isocyanic acids photochemically formed in the middle and
upper parts of the boundary layer were important sources for those at ground level in
urban region. The differences of the diurnal patterns between CICs and ground-level
concentrations of formic and isocyanic acids further highlight the importance of vertical
observations in elucidating their budgets and sources in the whole boundary layer.

Characterization of the vertical variations in formic and isocyanic acids could

provide valuable information for elucidating their budgets and sources in the boundary



layer. However, there are still many important but unresolved questions associated with
the vertical distributions of formic and isocyanic acids. For example, the key precursors
that drive the rapid formation of formic and isocyanic acids in the residual layer are still
unknown. Are there any changes in the key precursors and formation pathways of
formic and isocyanic acids with the increase of height in urban region? To answer these
questions, the combination of vertical gradient measurements of more chemical species
and numerical simulations is needed in future studies.
**Supporting Information:** Additional experimental details, materials, and methods,
including schematic illustration of tubing test, determination of the long tubes'
cumulative influence, and calculation of CICs..
## Data availability
Data related to this article are available online
at https://doi.org/10.7910/DVN/ANH0WE.
## Author contributions
QY, XBL, BY, and YH designed the research. QY, XBL, BY, XZ, YH, LY, XH,
JQ and MS contributed to the data collection and data analysis. QY and XBL wrote the
paper with contributions from all coauthors. All the coauthors discussed the results and
reviewed the paper.
## Competing interests
The authors declare that they have no conflict of interest.
## Acknowledgment
This work was financially supported by the National Key R&D Plan of China
(grant No. 2023YFC3706103, 2023YFC3706201, 2022YFC3700604) and the National
Natural Science Foundation of China (grant No. 42121004, 42275103, 42230701,
42305095). This work was also supported by the Special Fund Project for Science and



Technology Innovation Strategy of Guangdong Province (Grant No. 2019B121205004).
The authors would like to thank the personnel who participated in data collection,
instrument maintenance, and logistic support during the field campaign.

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



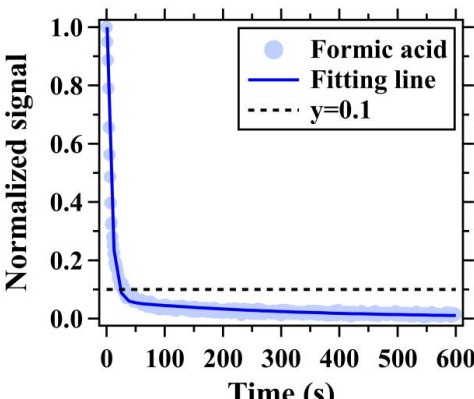


**Figure 1.** Depassivation curves of formic acid signal measured by I⁻ ToF-CIMS for the

400 m long tubing at the flow rate of 13 SLPM. Ion signals were normalized to those

measured at the start time (0 s) of the step-function change.

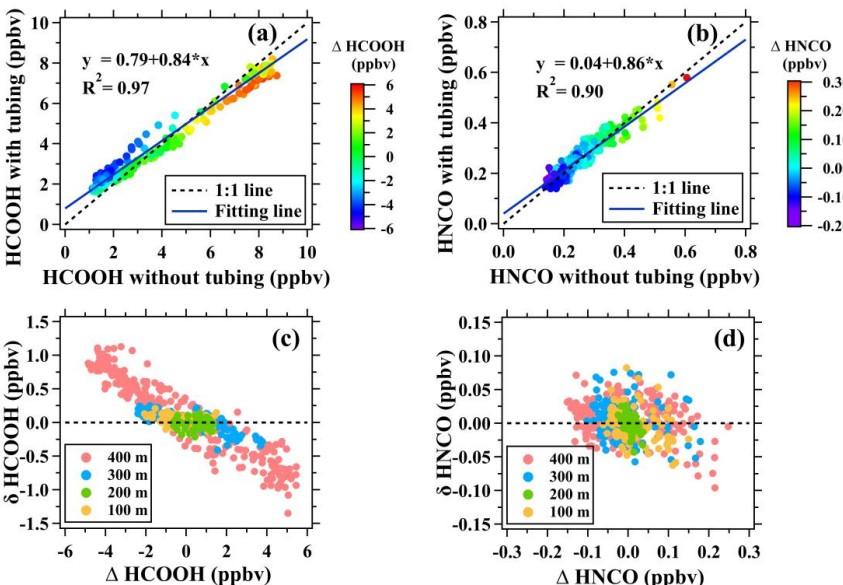

**Figure 2.** Assessment of long tubes in measuring formic and isocyanic acids in ambient air. (a-b) Scatterplots of mixing ratios of formic and isocyanic acids measured with the 400 m long tube versus those measured without the long tube. (c-d) Scatterplots of $\Delta[HCOOH]$ versus $\delta[HCOOH]$ and scatterplots of $\Delta[HNCO]$ versus $\delta[HNCO]$ for the 100, 200, 300, and 400 m tubes.



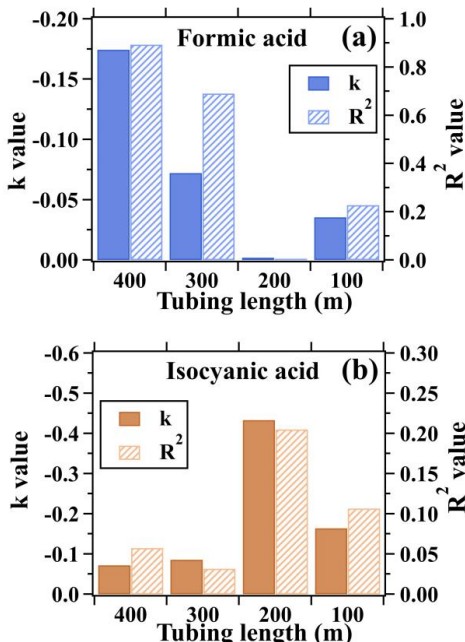

1006

**Figure 3.** Linear fitting parameters (namely $k$ and $R^2$) for (a) $\Delta[HCOOH]$ versus $\delta[HCOOH]$ and (b) $\Delta[HNCO]$ versus $\delta[HNCO]$. The scatterplots are shown in Figure 2. $k$ and $R^2$ are the slope and determination coefficient of the linear fitting lines, respectively.



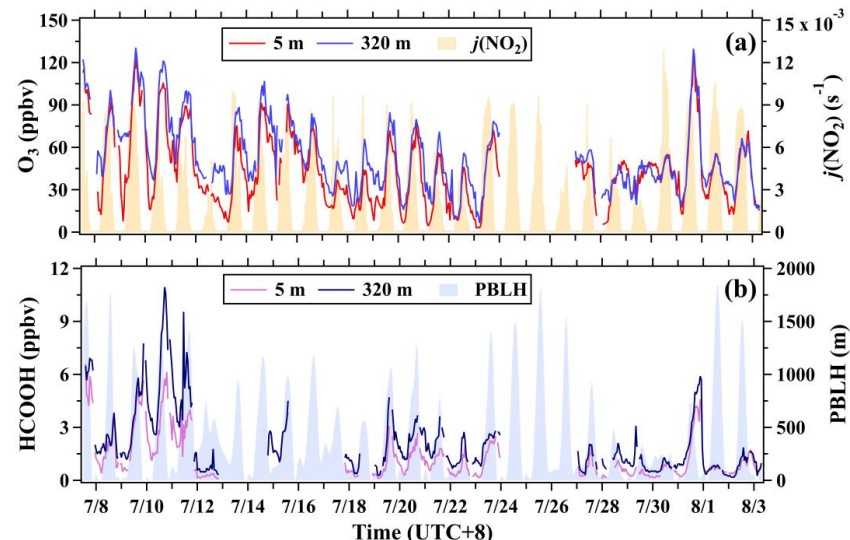

**Figure 4.** Time series of (a) $O_3$ (5 and 320 m), $j(NO_2)$, (b) formic acid (5 and 320 m),

and planetary boundary layer height (PBLH) during the campaign.



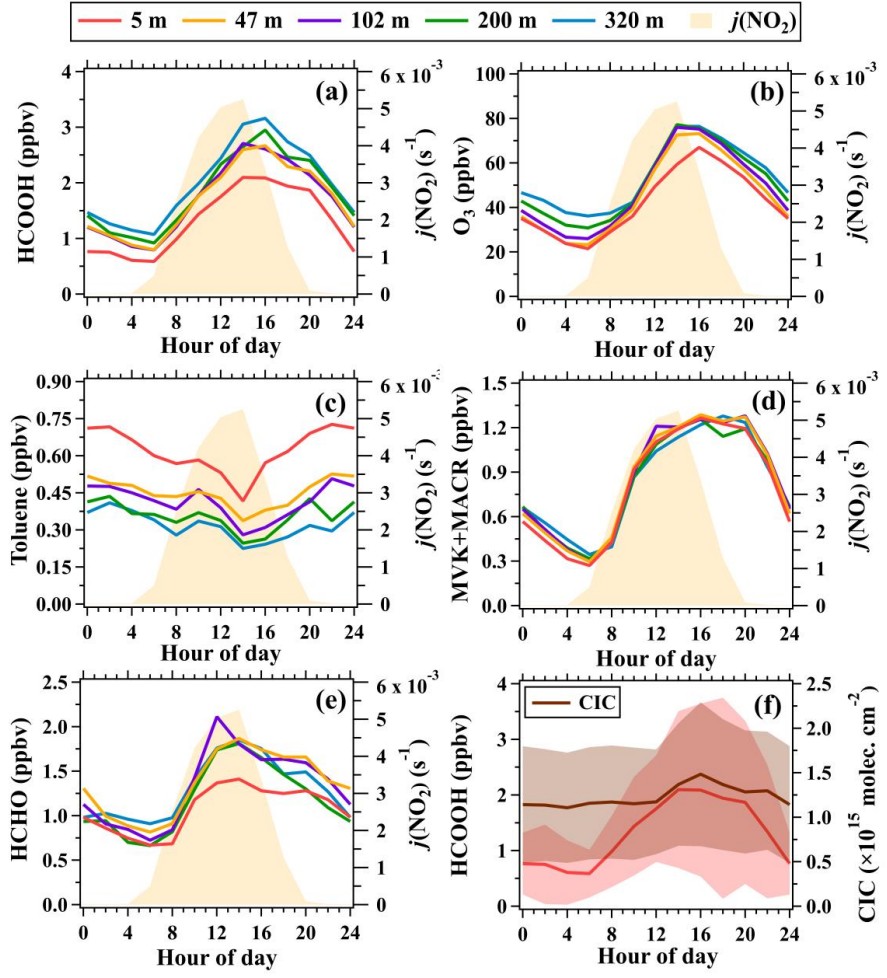

1014

**Figure 5.** Average diurnal variations in mixing ratios of (a) formic acid, (b) O₃, (c)

toluene, (d) MVK+MACR and (e) formaldehyde at 5, 47, 102, 200, and 320 m. (f)

Average diurnal variations in mixing ratios (5 m) and CICs of formic acid during the

field campaign; The shaded areas in panel (e) are half of the standard deviations.





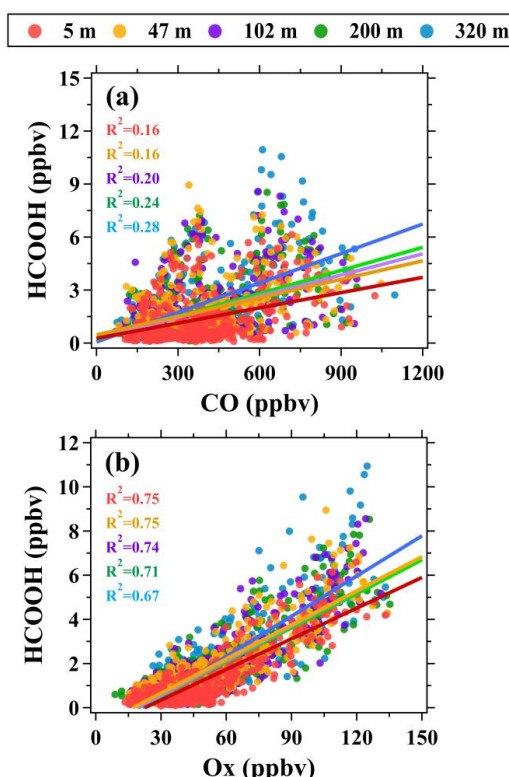


**Figure 6.** Scatter plots of (a) formic acid versus CO and (b) formic acid versus Ox at

different altitudes during the campaign.

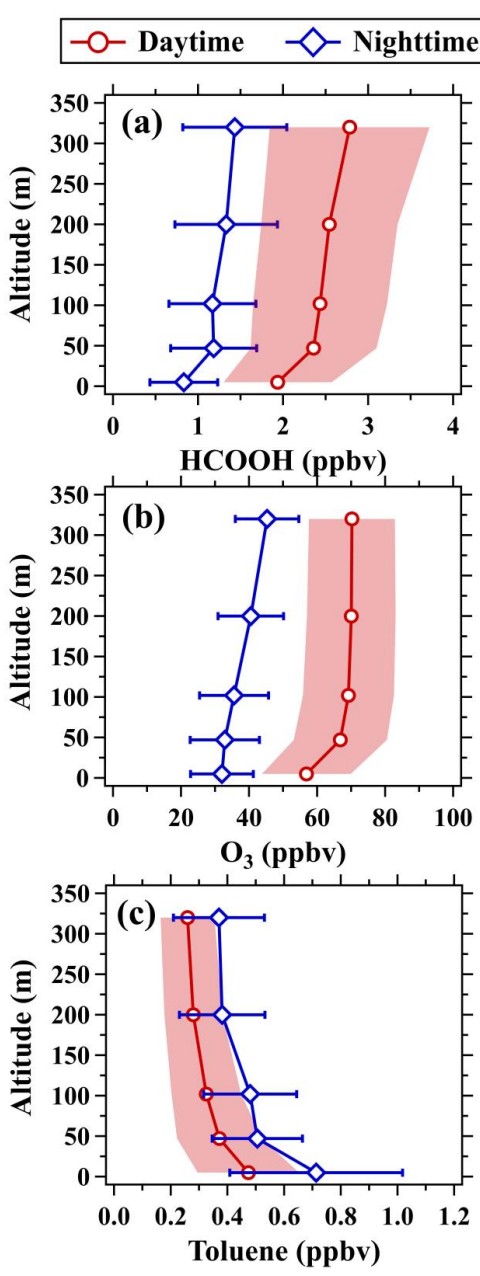


**Figure 7.** Vertical profiles of (a) formic acid, (b) $O_3$, and (c) toluene in daytime (11:00-

16:00 LT) and nighttime (22:00-5:00 LT). The shaded areas and error bars are half of

the standard deviations.



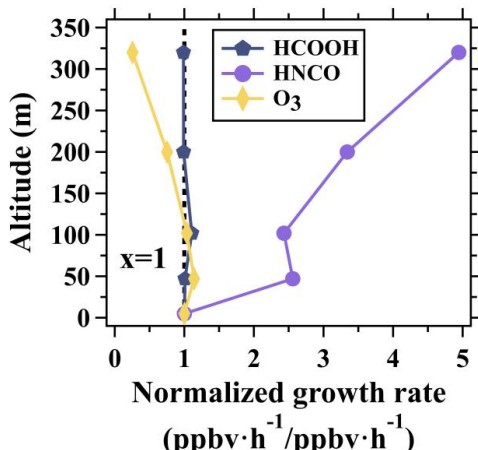


**Figure 8.** Normalized vertical profiles of the growth rates of ozone, formic acid, and

isocyanic acid between 6:00-10:00 LT averaged over the whole campaign. Growth rates

of the species at different altitudes were normalized to those at 5 m.



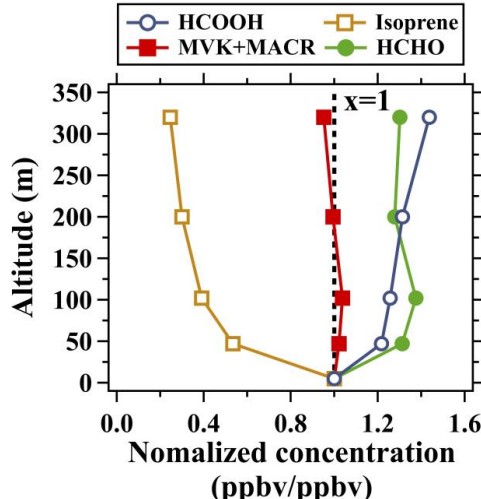


**Figure 9.** Normalized vertical profiles of formic acid, isoprene, formaldehyde, MVK
and MACR in daytime (11:00-16:00 LT) averaged over the whole campaign. Mixing
ratios of the species at different altitudes were normalized to those at 5 m.

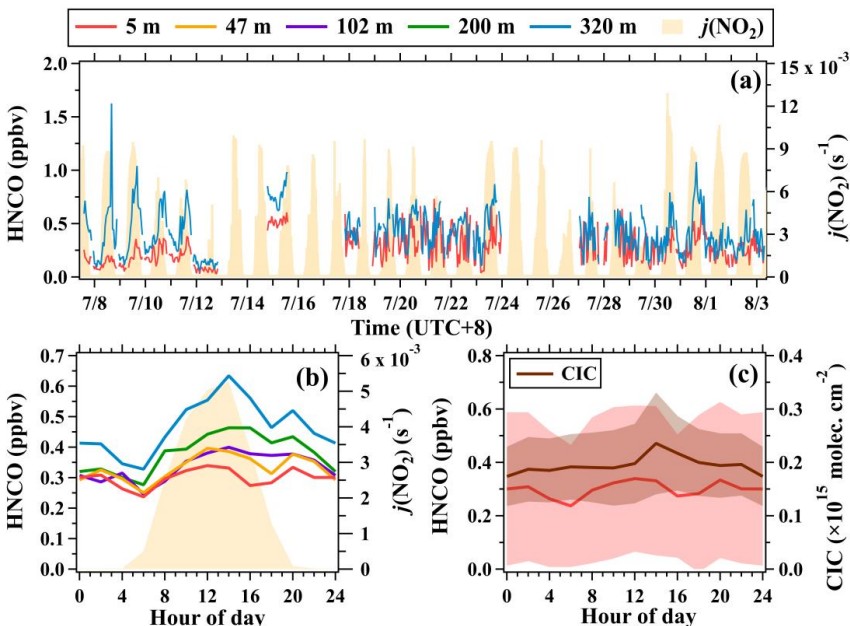

**Figure 10.** (a) Time series of isocyanic acid (5 and 320 m) and $j(NO_2)$. (b) Average diurnal variations in isocyanic acid at 5, 47, 102, 200, and 320 m. (c) Average diurnal variations in mixing ratios (5 m) and CICs of isocyanic acid during the campaign; The shaded areas in panel (c) are half of the standard deviations.

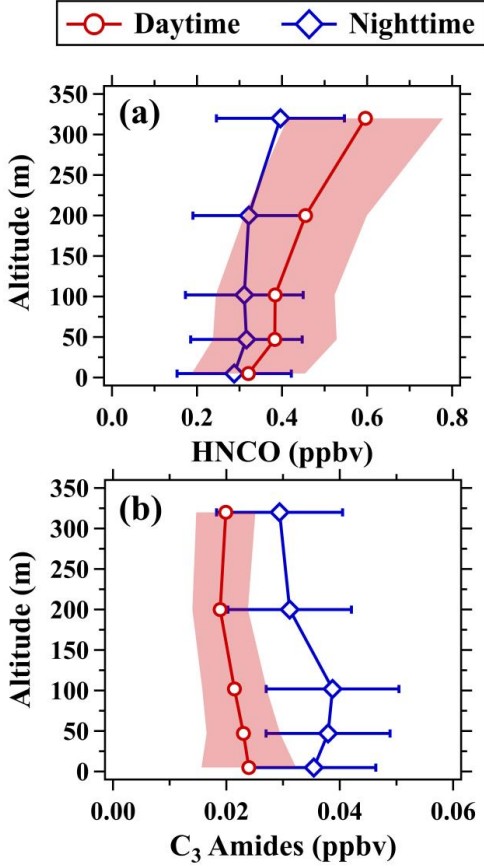


**Figure 11.** Vertical profiles of (a) isocyanic acid and (b) C$_3$ amides in daytime (11:00-
16:00 LT) and nighttime (22:00-5:00 LT). The shaded areas and error bars are half of
the standard deviations.

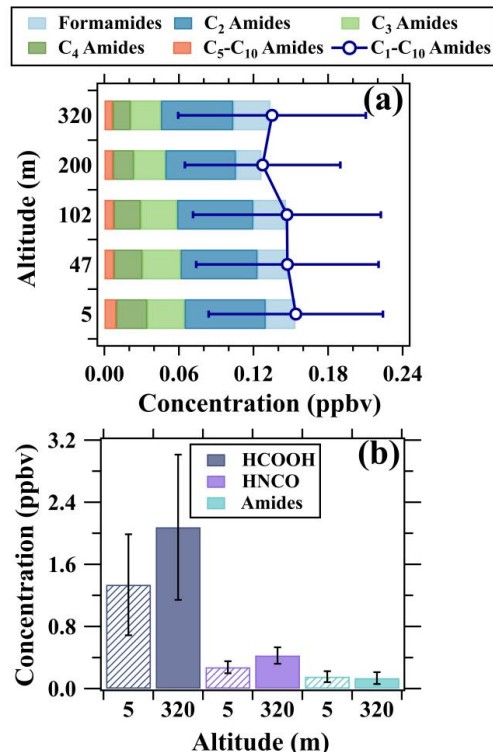


**Figure 12.** (a) Vertical variations in composition and concentrations of amides. (b) Concentration comparison of formic acid, isocyanic acid, and amides between 5 and 320 m. The data in both (a) and (b) was the average results of the whole campaign.