# Peer review of "Measurement report: Enhanced photochemical formation of"

_EGUsphere, 2024_

## Referee Comment (RC1)

**Review of** "Measurement report: Enhanced photochemical formation of formic and isocyanic acids in urban region aloft: insights from tower-based online gradient measurements"

**General Comments**

In this manuscript the authors present interesting measurements of a variety of VOCs along the vertical gradient of a very tall tower in Beijing, China. They focus on measurements of formic acid and isocyanic acid and attempt to explain how photochemical processes could be contributing to the variation in measured vertical gradients. I think this is an interesting set of measurements and worth publication. I left some comments that hopefully can improve the presentation and communication of the results.

**Specific Comments**

**(Introduction)** A reference that is missing that I think is relevant is Alwe, et al. (2019).[1] The authors of that study took measurements of formic acid eddy-covariance fluxes and vertical gradients in a forest and is one of the few studies of vertical gradient measurements of formic acid.

**(Line 78-81)** I believe that the 1 ppb number comes from Roberts, et al. (2011) and is based off a calculation where the concentration of NCO- ions were measured from cells *in vitro* that had undergone carbamylation and based on the henry's law constant of HNCO the 1 ppb threshold for potential health impacts was derived. The Fulgham, et al. 2020 study did not establish any threshold for acceptable levels of ambient HNCO so I recommend the authors remove the citations for the two studies associated with this statement. They could instead cite Roberts, et al. (2011).

**(Line 151)** What was the pore size and diameter of the filters?

**(Line 169-188)** The information related to the Figaero in this paragraph is confusing. If the authors are doing 4-min interval switching between the different sampling heights how does that impact the Figaero measurements? Were these measurements done the whole campaign? Did the authors just stop on one location for an hour and sample particles? I looked briefly ahead to see where Figaero measurements were discussed, but it's not clear. If these measurements aren't discussed can they be removed from this section?

**(Line 204)** I think this refers to "high" mass spectral peak resolution, not time.

**(Line 210)** Did the trace gas instrumentation have trouble maintaining adequate sample pressure when sampling through those long lines with the high flowrates?

**(Line 250-252)** HNCO is not an inorganic acid, it is an organic acid. The justification provided by the authors here is not valid. The authors should instead simply state that tubing delays are not discussed for HNCO.

**(Figure 1)** Can the authors include a vertical line in the plot that intersects with the formic acid trace to graphically show the determined delay time of 23 s?

**(Line 273)** I might be unfamiliar with this Taylor dispersion calculation. How is this calculation different from a plug-flow residence time calculation? For 13 SLPM, 0.37" inner diameter tube, and 400 m I calculated a plug-flow residence time of 129 s. Why is the Taylor dispersion number so low? Can the authors explain how to reconcile my plug-flow residence time calculation with their measured delay time of 29 s?

(**Line 273 or Section 2.1**) Can the authors report plug-flow residence times for the different lengths of tubing?

(**Line 292-303**) Could the authors decrease the influence of the long tailing by having a bypass pump that passivated the long lines while they were not being sampled?

(**Section 3.1**) I felt maybe this section could be condensed considerably. The main points are that the authors didn't observe any major impacts on tubing length when measuring HNCO and they were able to constrain tubing delay effects for HCOOH at the 400 m tubing length. I don't understand the calculation that was performed for Figure S3 and I don't understand what the authors are trying to convey with it. What is the significance of the $\Delta t = 14$ h for HCOOH? Is this saying that it takes 14 hours before the 400 m tubing is passivated?

(**Figure 3**) The authors conclude that HNCO does not experience the same tubing effects that HCOOH does. Is it useful to show the plot for HNCO then? Panel d in Figure 2 does not seem to imply a linear relationship and thus these data are confusing to show. Figure 3a is also only referred to briefly and it's not clear what the physical meaning of the slope is for the fit of $\delta$HCOOH and $\Delta$HCOOH. I think the authors try and explain this analysis some in Lines 317-319, but I'm still not very clear on what specific information this analysis is providing. I'm not sure why the $R^2$ is shown in Figure 3 either, but after looking at it I wonder why the 200 m length tubing seems to be an outlier in this pattern.

(**Line 353-361**) Since the source of formic acid here is considered to originate from photochemistry, would it be more interesting to compare median formic acid mixing ratios measured between 10:00 and 15:00 between the two measurement heights? I wonder if a lot of variability is being introduced into the averages because HCOOH is normally high during the day and low at night. That comparison may show an even more compelling difference between the 5 m and 320 m heights.

(**Figure 5**) Figure 5 is a little confusing because CICs are shown in panel f but Figures 6-9 are discussed in the text before CICs are introduced on Line 466.

(**Line 408**) I find the "growth rate" terminology unusual. It immediately makes me think of new particle formation. Would the authors consider rephrasing "growth rate" to "enhancement rate" or "gradient"?

(**Line 408**) How were the "growth rates" calculated?

(**Figure 8**) Instead of the x-axis being "growth rate" could it just be normalized concentration like Figure 9?

(**Line 466**) Maybe the authors should just say boundary layer instead of mentioning the residual layer.

(**Line 466**) My intuition says that an atmospheric column integration calculation would be concentration times the assumed volume of the column (which should only vary as a function of boundary layer height). I'm having a hard time conceptualizing molecules cm$^{-2}$.

(**Figure 5f**) So does this mean that the total number of molecules throughout the boundary layer stay constant as a function of hour of day? Can the authors explain this more in the text? I'm not sure I connect how this calculation demonstrates how ground measurements are not capturing vertical distribution dynamics of HCOOH better than the gradient measurements shown in other figures.

(**Line 543 and Figure 12**) It's really interesting to see measurements of all the different amides as a function of height of the tower. On the other hand, the authors seem to be reaching further than the data suggest to explain small differences in concentrations with a complicated chemical mechanism that was

vaguely speculated around Line 100 in the introduction. It's okay if the authors can't explain where these chemicals come from. The measurements are fairly novel and interesting in and of themselves. A much more detailed chemical analysis is warranted to articulate chemical mechanisms responsible for HNCO production.

**(Line 543)** Did the amides experience any tubing effects?

**(Line 564)** Please replace all mentions of "growth rates" with more suitable terminology like suggested above.

**(Line 574)** Again, I don't really understand the utility of using the CIC calculation to critique ground level measurements. If the authors want to make the point that chemicals can be produced at higher altitudes and so using ground level measurements for modeling may not be adequate for answering some questions that would be reasonable.

**(Conclusions)** I'm confused about the authors conclusions regarding the tubing analysis. My understanding from the data is that formic acid experiences significant tubing effects from the long lines and HNCO did not. The authors didn't really provide any information on if they had to correct the data for tubing delays or best practices when measuring formic acid from really long lines. Did the authors have to correct HCOOH concentrations for the tubing effects?

**Technical Corrections**

**(Line 72)** HNCO is an organic acid.

**(Line 98-102)** Can the authors start this hypothesized mechanism with "We hypothesize.." or something like that to indicate that this potentially important mechanism is the authors thoughts?

**(Line 93-105)** The authors might consider removing this paragraph. There's three sentences of loose speculation in it and I think the important point could be summarized in a single sentence tacked on to the last sentence of the previous paragraph. Basically "vertical gradient measurements of HNCO can help elucidate potential formation sources and mechanisms".

**(Line 106)** Replace "mass spectrometer" with "mass spectrometry"

**(Line 108-110)** I would recommend removing this statement. In the following sentences the authors cite studies that provide evidence to the contrary. In 2013 a group from Cal-Tech put their CIMS instrument on top of a tower for measurements.

**(Line 125)** Remove "alkane" from PFA definition.

**(Line 172)** Replace "ionic" with "ion molecule reaction"

**(Line 175)** "MIR" should be "IMR"

**(Line 179-181)** Move this sentence to Line 170 in front of "A filter inlet for gases and aerosols…"

**(Line 201)** Replace "eliminated" with "corrected for"

**(Line 366-369)** Consider rewording…

"Photochemical formation of secondary pollutants was largely suppressed from July 13th to 30th due to weak solar radiation (characterized by small $j(NO_2)$ values) and precipitation."

**(Figure 8)** Instead of putting X=1 in the figure can the authors just write in the caption that a dotted line is shown for the normalized "growth rate" of 1?

**(Line 436)** Consider rephrasing to "Our results point to the likely importance of photochemistry as a source of formic acid that is enhanced with increasing height within the boundary layer."

**(Line 445)** Consider citing Link, et al. (2021).[2]

**(Line 470)** I think the authors mean Figure 5f.

**(Figure 12)** Can the authors provide a description in the caption of what the patterning indicates for the bars?

**(Barnes, et al. 2010 Reference)** Should the journal abbreviation be ChemPhyChem?

**References for Review**

(1) Alwe, H. D.; Millet, D. B.; Chen, X.; Raff, J. D.; Payne, Z. C.; Fledderman, K. Oxidation of volatile organic compounds as the major source of formic acid in a mixed forest canopy. *Geophysical research letters* **2019**, *46* (5), 2940-2948.
(2) Link, M. F.; Brophy, P.; Fulgham, S. R.; Murschell, T.; Farmer, D. K. Isoprene versus Monoterpenes as Gas-Phase Organic Acid Precursors in the Atmosphere. *ACS Earth and Space Chemistry* **2021**, *5* (6), 1600-1612.

---

## Author Comment (AC1)

* * *
**Response to Reviewer #1**
* * *
**1.** In this manuscript the authors present interesting measurements of a variety of VOCs along the vertical gradient of a very tall tower in Beijing, China. They focus on measurements of formic acid and isocyanic acid and attempt to explain how photochemical processes could be contributing to the variation in measured vertical gradients. I think this is an interesting set of measurements and worth publication. I left some comments that hopefully can improve the presentation and communication of the results.

**Reply:** We appreciate your valuable comments and suggestions, which are very helpful for the improvement of our manuscript.

**2.** (Introduction) A reference that is missing that I think is relevant is Alwen, et al. (2019).1 The authors of that study took measurements of formic acid eddy-covariance fluxes and vertical gradients in a forest and is one of the few studies of vertical gradient measurements of formic acid.

**Reply:** We greatly appreciate your recommendation. Alwe et al. used PTR-QiTOF-MS to make vertical gradient measurements of formic acid in a forest area at six heights (namely 5, 9, 13, 17, 21, and 34 m) on the PROPHET tower and emission fluxes using the eddy covariance method. They found that formic acid is predominantly contributed by secondary formation rather than primary emission. The measurement techniques used in our study are analogous to those used in the literature, but we used longer tubes to obtain vertical gradients of formic acid within a higher altitude range. Results of the two studies also confirmed each other. This article has been cited in the introduction of our manuscript. [see P: 4-5; L:69-72]

*"In combination with vertical gradient and flux measurements of formic acid in a forest ecosystem, Alwe et al. (2019) suggested that secondary formation, rather than primary emission, is the major source of ambient formic acid."*

**3.** (Line 78-81) I believe that the 1 ppb number comes from Roberts, et al. (2011) and is based off a calculation where the concentration of NCO- ions were measured from cells in vitro that had undergone carbamylation and based on the henry's law constant of HNCO the 1 ppb threshold for potential health impacts was derived. The Fulgham, et al. 2020 study did not establish any threshold for acceptable levels of ambient HNCO so I recommend the authors remove the citations for the two studies associated with this statement. They could instead cite Roberts, et al. (2011).

**Reply:** We appreciate your valuable comments and suggestions. The threshold of 1 ppbv for ambient HNCO concentration was calculated by Robert et al. based on an effective Henry's Law solubility of $10^5$ M/atm at pH=7.4, with an NCO$^-$ concentration of 100 μM in vitro experiments. Here we want to emphasize that HNCO will induce the protein carbamylation process in human body and impose severe threats to human health. Verbrugge et al. have detailed the sources of HNCO in human body, the mechanism of HNCO causing carbamylation, and the possible effects of the protein carbamylation process. We have rewritten this sentence in the manuscript to make it clearer. [see P: 5; L: 83-86]

*"The mixing ratio of HNCO in the atmosphere exceeding 1 ppbv may endanger human health (Roberts et al., 2011), and the protein carbamylation caused by HNCO in human body may induce various risks (Verbrugge et al., 2015)."*

**4.** (Line 151) What was the pore size and diameter of the filters?

**Reply:** We appreciate your valuable comments. The diameter of the filter (Whatman) is 46.2 mm and the pore size is 2 μm. We have rephrased this sentence in the revised manuscript. [see P: 7; L: 147-152]

*"After removing fine particles by PFA Teflon filters (Whatman) with diameters of 46.2 mm and pore sizes of 2 μm, ambient air at four altitudes on the tower (namely 47, 102, 200, and 320 m) was simultaneously and continuously drawn to the ground through long PFA Teflon tubes (100, 150, 250, and 400 m; outer diameter: 1/2"; inner diameter: 0.374") by a vacuum pump."*

**5.** (Line 169-188) The information related to the Figaero in this paragraph is confusing. If the authors are doing 4-min interval switching between the different sampling heights how does that impact the Figaero measurements? Were these measurements done the whole campaign? Did the authors just stop on one location for an hour and sample particles? I looked briefly ahead to see where Figaero measurements were discussed, but it's not clear. If these measurements aren't discussed can they be removed from this section? [see P: 7-8; L: 147-152]

**Reply:** We appreciate your valuable comments and suggestions. During the field campaign, both gaseous and particle measurements were made through the FIGAERO of the CIMS, but only gaseous measurements were discussed in this manuscript.

In a one-hour cycle, the first 24 min was allocated to make gaseous measurements during which a complete vertical profile of a gaseous species can be obtained. In the gaseous measurement mode, a rapid blank measurement was made for 10 s at 3-min intervals in the first 21 min and a long-time blank measurement was made in the rest 3 min. During the first 21-min period of the one-hour cycle, another inlet at 5 m was used to collect ambient particles using PTFE membrane filters (Zefluor®, Pall Inc., USA). Therefore, the remaining 36 min of the one-hour cycle was allocated to analyze the collected particle. We have provided more description for the operation setup of the CIMS instrument in the manuscript. [see P: 8-9; L: 181-190]

*"During the field campaign, both gaseous and particle measurements were made through the FIGAERO of the CIMS, but only gaseous measurements were analyzed in this study. In a one-hour cycle, the first 24 min was allocated to make gaseous measurements during which a complete vertical profile of a gaseous species can be obtained. In the gaseous measurement mode, a rapid blank measurement was made for 10 s at 3-min intervals in the first 21 min and a long-time blank measurement was made in the rest 3 min (Palm et al., 2019). During the first 21-min period of the one-hour cycle, another inlet at 5 m was used to collect ambient particles using PTFE membrane filters (Zefluor®, Pall Inc., USA). Therefore, the remaining 36 min of the one-hour cycle was allocated to analyze the collected particle."*

**6.** (Line 204) I think this refers to "high" mass spectral peak resolution, not time.

**Reply:** Thanks for pointing out this mistake and we have corrected it throughout the manuscript. [see P: 9; L: 206-209]

*"A high-resolution proton-transfer-reaction quadrupole interface time-of-flight mass spectrometry (PTR-ToF-MS) with both $H_3O^+$ and $NO^+$ ion chemistry was used to measure reported precursors of the two acids, such as isoprene, aromatics, OVOCs, and amides."*

**7.** (Line 210) Did the trace gas instrumentation have trouble maintaining adequate sample pressure when sampling through those long lines with the high flowrates?

**Reply:** We appreciate your valuable comments and suggestions. Before the campaign, we conducted a series of tests both in our lab and at a tower site in Shenzhen, China to determine the suitable range of the flow rates in these long tubes that can largely reduce tubing delays, as well as maintain adequate sample pressure for the instruments. We found that the pressure drop was ~12% even in the 400 m long tubing at the flow rate of ~13 SLPM. In addition, the detection-cell pressures of the instrument were checked and recorded every day during the field campaign to ensure that the gradient measurements were valid. A detailed introduction to the tubing assessment and the observation system establishment has been reported in our previous study *(Li et al., 2023)*.

*Reference:*

*Li, X.-B., Zhang, C., Liu, A., Yuan, B., Yang, H., Liu, C., Wang, S., Huangfu, Y., Qi, J., Liu, Z., He, X., Song, X., Chen, Y., Peng, Y., Zhang, X., Zheng, E., Yang, L., Yang, Q., Qin, G., Zhou, J., and Shao, M.: Assessment of long tubing in measuring atmospheric trace gases: applications on tall towers, Environmental Science: Atmospheres, 3, 506-520,10.1039/d2ea00110a 2023.*

**8.** (Line 250-252) HNCO is not an inorganic acid, it is an organic acid. The justification provided by the authors here is not valid. The authors should instead simply

state that tubing delays are not discussed for HNCO.

**Reply:** We appreciate your valuable comments and suggestions. We have conducted a more extensive investigation on the definition of HNCO. As one of the authoritative chemistry websites, ChemSpider provides the definition of isocyanic acid as "A colorless, volatile, poisonous inorganic compound with the formula HNCO" (https://www.chemspider.com/Chemical-Structure.6107.html?rid=5179dbb4-99aa-4c52-83eb-b01d47122023). In addition, HNCO is also recognized as an inorganic acid in the literature according to its chemical properties *(Roberts et al., 2010, 2011)*.

*References:*

*(1) Roberts, J. M., Veres, P. R., Cochran, A. K., Warneke, C., Burling, I. R., Yokelson, R. J., Lerner, B., Gilman, J. B., Kuster, W. C., Fall, R., and de Gouw, J.: Isocyanic acid in the atmosphere and its possible link to smoke-related health effects, Proceedings of the National Academy of Sciences, 108, 8966-8971, 10.1073/pnas.1103352108, 2011.*

*(2) Roberts, J. M., Veres, P., Warneke, C., Neuman, J. A., Washenfelder, R. A., Brown, S. S., Baasandorj, M., Burkholder, J. B., Burling, I. R., Johnson, T. J., Yokelson, R. J., and de Gouw, J.: Measurement of HONO, HNCO, and other inorganic acids by negative-ion proton-transfer chemical-ionization mass spectrometry (NI-PT-CIMS): application to biomass burning emissions, Atmospheric Measurement Techniques, 3, 981-990, 10.5194/amt-3-981-2010, 2010.*

**9.** (Figure 1) Can the authors include a vertical line in the plot that intersects with the formic acid trace to graphically show the determined delay time of 23 s?

**Reply:** We appreciate your valuable suggestion. We have added a vertical line in Figure 1 to clearly show how the delay time of 23 s for formic acid is determined.

[Figure]

**Figure 1.** *Depassivation curves of formic acid signal measured by I⁻ ToF-CIMS for the 400 m long tubing at the flow rate of 13 SLPM. Ion signals were normalized to those measured at the start time (0 s) of the step-function change.*

**10.** (Line 273) I might be unfamiliar with this Taylor dispersion calculation. How is this calculation different from a plug-flow residence time calculation? For 13 SLPM, 0.37" inner diameter tube, and 400 m I calculated a plug-flow residence time of 129 s. Why is the Taylor dispersion number so low? Can the authors explain how to reconcile my plug-flow residence time calculation with their measured delay time of 29 s?

   **Reply:** We appreciate your valuable comments and suggestions. We think the plug-flow residence time you mentioned refers to the time required for the sample gas to pass through the tubes and can be calculated using Eq. (1). Theoretically, the residence time is ~131 s in a 400 m long tubing at a flow rate of 13 SLPM. The difference between residence time and delay time is discussed in detail in our previous work *(Li et al., 2023)*.

$$t = \frac{L\pi \left(\frac{d}{2}\right)^2}{Q} \tag{1}$$

where $t$ is the residence time (s), $L$ is the tubing length (cm), $d$ is the inner diameter of the tubing (cm), $Q$ is the flow rate of the sample gas stream in tubing (cm⁻³ s⁻¹).

   As for the measured tubing delays of trace gases, they refer to the amounts of time

required for the instruments to measure stable concentrations of targeted species in response to a change in species concentrations at the tubing inlet. This definition has been mentioned in the Methods and materials and used in the literature *(Pagonis et al., 2017; Liu et al., 2019)*. As also recommended in the literature, the tubing delay of a trace gas is usually defined as the amount of time required to reach 90% of the change in its concentration before entering the tubing.

According to the method used in the work by Karion et al., *(2010)*, the influence times of molecular diffusion and dispersion, denoted by $t_m$ (s), on measured concentrations of trace gases after traversing a tubing can be estimated using Eq. (2):

$$t_m = \frac{X_{eff}}{\bar{V}} \tag{2}$$

where $\bar{V}$ is the average flow velocity (cm s⁻¹) in tubing, $X_{eff}$ is the effective distance (cm) of molecular dispersion, namely the longitudinal mixing length of molecules driven by molecular diffusion and Taylor dispersion; $X_{eff}$ can be estimated using Eq. (3):

$$X_{eff} = \sqrt{(2D_{eff}t)} \tag{3}$$

where $t$ (s) is the residence time of the air sample in tubing, $D_{eff}$ is the effective molecular diffusion coefficient (cm² s⁻¹) and can be estimated using Eq. (4):

$$D_{eff} = D + \frac{a^2\bar{V}^2}{48D} \tag{4}$$

where $D$ is the molecular diffusion coefficient (cm² s⁻¹) of the trace gas in air and $a$ is the inner radius (cm) of the tubing.

Taylor dispersion is caused by the combined effect of molecular diffusion and mechanical dispersion. According to our calculation, the effect of Taylor dispersion on the tubing delay of trace gases is relatively small and can be neglected.

**References:**

*(1) Li, X., Zhang, C., Liu, A., Yuan, B., Yang, H., Liu, C., Wang, S., Huangfu, Y., Qi, J.,*

*Liu, Z., He, X., Song, X., Chen, Y., Peng, Y., Zhang, X., Zheng, E., Yang, L., Yang, Q., Qin, G., Zhou, J., and Shao, M.: Emerging investigator series: Assessment of Long Tubing in Measuring Atmospheric Trace Gases: Applications on Tall Towers, Environmental Science: Atmospheres, 10.1039/d2ea00110a, 2023.*

*(2) Pagonis, D., Krechmer, J. E., de Gouw, J., Jimenez, J. L., and Ziemann, P. J.: Effects of gas–wall partitioning in Teflon tubing and instrumentation on time-resolved measurements of gas-phase organic compounds, Atmospheric Measurement Techniques, 10, 4687-4696, 10.5194/amt-10-4687-2017, 2017.*

*(3) Liu, X., Deming, B., Pagonis, D., Day, D. A., Palm, B. B., Talukdar, R., Roberts, J. M., Veres, P. R., Krechmer, J. E., Thornton, J. A., de Gouw, J. A., Ziemann, P. J., and Jimenez, J. L.: Effects of gas–wall interactions on measurements of semivolatile compounds and small polar molecules, Atmospheric Measurement Techniques, 12, 3137-3149, 10.5194/amt-12-3137-2019, 2019.*

*(4) Karion, A., Sweeney, C., Tans, P., and Newberger, T.: AirCore: An Innovative Atmospheric Sampling System, Journal of Atmospheric and Oceanic Technology, 27, 1839-1853, 10.1175/2010jtecha1448.1, 2010.*

**11.** (Line 273 or Section 2.1) Can the authors report plug-flow residence times for the different lengths of tubing?

**Reply:** We appreciate your valuable suggestion. We have calculated the residence times for different lengths of tubes based on the average flow rate in each tubing measured during the field campaign (Table S1). The results have been provided in Table S1 in SI.

*Table S1. Flow rates and residence times of the different lengths of tubes during the field campaign*

| Length of tubing (m) | Altitude (m) | Flow rate (SLPM) | Calculated residence time (s) |
|---|---|---|---|
| ~4 | 5 | 20.4 | 0.8 |
| 100 | 47 | 17.7 | 24.0 |
| 150 | 102 | 17.8 | 35.8 |
| 250 | 200 | 15.5 | 68.6 |
| 400 | 320 | 13.6 | 125.0 |

**12.** (Line 292-303) Could the authors decrease the influence of the long tailing by having a bypass pump that passivated the long lines while they were not being sampled?

**Reply:** We appreciate your valuable suggestions. During the field campaign, a vacuum pump was used as a bypass pump to draw all the five tubes continuously and simultaneously with flow rates ranging between 13 and 21 SLPM (Table S1). Therefore, this vacuum pump provided large flow rates to passivate the long lines when they were not sampled. A schematic illustration of the vertical observation system (Figure S1) was provided in SI.

[Figure]

**Figure S1.** *A simple schematic illustration of the vertical observation system on the Tower and locations of the five sampling inlets for measuring atmospheric gaseous species.*

**13.** (Section 3.1) I felt maybe this section could be condensed considerably. The main points are that the authors didn't observe any major impacts on tubing length when measuring HNCO and they were able to constrain tubing delay effects for HCOOH at the 400 m tubing length. I don't understand the calculation that was performed for Figure S3 and I don't understand what the authors are trying to convey with it. What is the significance of the Δt = 14 h for HCOOH? Is this saying that it takes 14 hours before the 400 m tubing is passivated?

**Reply:** We appreciate your valuable comments and suggestions. In ambient environment tests, we observed that the usage of the 400 m long tubing resulted in an underestimation of ambient concentrations of formic and isocyanic acids. In this condition, we want to clarify the reasons for these measurement uncertainties and quantify its impact on the measurements. The same assessment method for isocyanic acid as that for formic acid could not only demonstrate the reliability of the method but

also further verify the feasibility of using long tubing for isocyanic acid measurement.

Based on the phenomenon that formic acid was underestimated during the day and overestimated at night when being measured through the long tubes. We speculate that these measurement uncertainties may be caused by the influence of long-tail effects. However, ambient concentrations of the trace gases changed with time and the long-tail effect also accumulated. For example, we cannot simply conclude that the underestimation of formic acid concentrations measured through the 400 m long tube at 12:00 LT was only caused by the increase in its ambient concentrations at 10:00 LT. They may also be related to the change in its ambient concentrations in a longer time ago. In this condition, we have tried to determine this time interval, namely $\Delta t$ in the manuscript. The $R^2$ between the average concentration change $\Delta HCOOH$ and the impact of the long tubing on the measurement $\delta HCOOH$ could reach 0.89 when $\Delta t$ equals 14 h. It indicates that the continuous concentration changes in the previous 14 h were the main reason for the underestimation of formic acid concentrations when measured through the 400 m long tube.

**14.** (Figure 3) The authors conclude that HNCO does not experience the same tubing effects that HCOOH does. Is it useful to show the plot for HNCO then? Panel d in Figure 2 does not seem to imply a linear relationship and thus these data are confusing to show. Figure 3a is also only referred to briefly and it's not clear what the physical meaning of the slope is for the fit of $\delta HCOOH$ and $\Delta HCOOH$. I think the authors try and explain this analysis some in Lines 317-319, but I'm still not very clear on what specific information this analysis is providing. I'm not sure why the R2 is shown in Figure 3 either, but after looking at it I wonder why the 200 m length tubing seems to be an outlier in this pattern.

**Reply:** We appreciate your valuable comments and suggestions. The tubing effects of isocyanic acid can be compared with those of formic acid, confirming the feasibility of the assessment method and demonstrating that isocyanic acid can be well measured using long tubes. However, the scatter plot of isocyanic acid in Figure 2(d) does not exhibit a linear correlation but fluctuates around the center at x=0 and y=0, suggesting

that isocyanic acid is almost unaffected by the long-tail effects even in different lengths of tubing.

As we stated in the results, $\delta$HCOOH is defined the difference between the mixing ratios of formic acid measured with and without the long tubing, which represents the impact of using a long tubing on measurements. $\Delta$HCOOH is defined as the change in mixing ratios of formic acid measured using long tubes at time $t$ relative to its average mixing ratio over the previous time interval of $\Delta t$, reflecting the variation in measured concentrations of the species. Therefore, the slope of $\delta$HCOOH and $\Delta$HCOOH reflects the underestimation or overestimation of the measured concentrations of the species caused by changes in its ambient concentrations. For example, if $k = -0.17$, an increase in the ambient concentration of 1 ppbv will result in an underestimation of the concentration of formic acid by 0.17 ppbv. The $R^2$ value can reflect the degree of influence that long-tail effects have on the measurements made through the long tubing. A higher $R^2$ value indicates a more significant impact.

During the test of the 200 m tubing, meteorological conditions significantly changed with lower temperatures and stronger winds in comparison to the days on which the tests of the other lengths of tubes were performed. As shown in Figure A1, the concentrations of formic acid and isocyanic acid were evidently enhanced and significantly varied during the 400 m tubing test. In contrast, ambient concentrations of formic and isocyanic acid were relatively low and slightly varied, resulting in the exceedingly large or low values of $k$ and $R^2$ between the concentrations of formic acid measured with and without the 200 m long tubing. However, according to the results of the test, the average concentration difference of formic and isocyanic acid measured with and without the 200 m tubing agreed well within 4%, suggesting that the 200 m long tube has minor effects on the measurements of formic and isocyanic acid.

[Figure]

**Figure A1.** *Time series of (a) formic and (b) isocyanic acid concentrations measured with and without the 400 m long tube.*

[Figure]

**Figure A2.** *Time series of (a) formic and (b) isocyanic acid concentrations measured*

*with and without the 200 m long tube.*

**15.** (Line 353-361) Since the source of formic acid here is considered to originate from photochemistry, would it be more interesting to compare median formic acid mixing ratios measured between 10:00 and 15:00 between the two measurement heights? I wonder if a lot of variability is being introduced into the averages because HCOOH is normally high during the day and low at night. That comparison may show an even more compelling difference between the 5 m and 320 m heights.

**Reply:** We appreciate your valuable suggestions. The median mixing ratios of formic acid at 5 m and 320 m between 10:00-15:00 LT were 1.5 ppbv and 2.3 ppbv, which were close to their average mixing ratios of 1.3 ppbv and 2.1ppbv. Similar to formic acid, the median mixing ratios of isocyanic acid between 10:00-15:00 LT were also close to their average mixing ratios. The median mixing ratios of isocyanic acid at 5 m and 320 m were 0.28 ppbv and 0.54 ppbv, respectively, and the average mixing ratios were 0.28 ppbv at 5 m and 0.43 ppbv at 320 m. Therefore, we think there is no significant difference between the usage of median and mean values.

**16.** (Figure 5) Figure 5 is a little confusing because CICs are shown in panel f but Figures 6-9 are discussed in the text before CICs are introduced on Line 466.

**Reply:** We appreciate your valuable suggestion. We have modified Figure 5 and adjusted Figure 5(f) to Figure 10. In addition, the average diurnal variation in the planetary boundary layer height (PBLH) was added in Figure 5.

[Figure]

***Figure 5.*** *Average diurnal variations in mixing ratios of (a) formic acid, (b) O₃, (c) toluene, (d) MVK+MACR, (e) formaldehyde at the five inlet heights and (f) PBLH and j(NO₂). The shaded areas in panel (f) are half of the standard deviations.*

[Figure]

***Figure 10.*** *Average diurnal variations in mixing ratios (5 m) and CICs of formic acid*

*during the field campaign; The shaded areas are half of the standard deviations.*

**17.** (Line 408) I find the "growth rate" terminology unusual. It immediately makes me think of new particle formation. Would the authors consider rephrasing "growth rate" to "enhancement rate" or "gradient"?

**Reply:** We appreciate your valuable suggestion and we have changed the terminology "growth rate" to "enhancement rate" in the manuscript.

**18.** (Line 408) How were the "growth rates" calculated?

**Reply:** We appreciate your valuable comments. The enhancement rate (namely the growth rate in the original manuscript) is defined as the average change rate of the species concentration between two adjacent hours. In this study, it is used to characterize the net photochemical formation rate of the species. We have defined the terminology "enhancement rate" in the revised manuscript. [see P: 17; L: 407-408]

*"The enhancement rate is defined as the average change rate of the species concentration between two adjacent hours."*

**19.** (Figure 8) Instead of the x-axis being "growth rate" could it just be normalized concentration like Figure 9?

**Reply:** We appreciate your valuable suggestion. In Figure 8, we want to express that isocyanic acid concentrations in the upper air increased more rapidly between 6:00-10:00 LT than those of formic acid and ozone. The formation of isocyanic acid aloft is significantly enhanced. Therefore, we think the x-axis being "enhancement rate" is more suitable.

**20.** (Line 466) Maybe the authors should just say boundary layer instead of mentioning the residual layer.

**Reply:** We appreciate your valuable comments and suggestions and we have provided a new figure (Figure 10) in the manuscript. The diurnal variation pattern and vertical profiles indicate that there was a pronounced secondary formation of formic and isocyanic acids during the daytime. In addition, concentrations of the two species

at the ground level may be affected by the entrainment of the residual layer with the growth of the boundary layer in daytime. Therefore, the boundary layer and the residual layer should be considered as a whole, when characterizing the abundance of formic and isocyanic acids in the atmosphere. So, we calculated the CICs in the boundary layer, as shown in Figure 10.

**21.** (Line 466) My intuition says that an atmospheric column integration calculation would be concentration times the assumed volume of the column (which should only vary as a function of boundary layer height). I'm having a hard time conceptualizing molecules cm-2.

**Reply:** We appreciate your valuable comments and agree with your opinion of the column integration calculation. In this study, the mixing ratios of the species (unit: ppbv) were converted to number concentrations (molecule/cm$^3$) first and then its CICs were calculated. The usage of CICs allows us to clearly demonstrate the variations in species abundance and their budgets in the PBL. In addition, the CICs of the species can be directly compared with remote sensing observations in the future.

**22.** (Figure 5f) So does this mean that the total number of molecules throughout the boundary layer stay constant as a function of hour of day? Can the authors explain this more in the text? I'm not sure I connect how this calculation demonstrates how ground measurements are not capturing vertical distribution dynamics of HCOOH better than the gradient measurements shown in other figures.

**Reply:** We appreciate your valuable comments. The stability of the column integration calculation does not imply that the number of molecules in the boundary layer remains constant. Instead, ground-level observations of the concentration are significantly affected by the development of the boundary layer, making it difficult to well characterize the change in the budget of formic acid. By contrast, the usage of CICs can largely reduce the effects of the PBL change by considering the vertical gradients of the species.

**23.** (Line 543 and Figure 12) It's really interesting to see measurements of all the

different amides as a function of height of the tower. On the other hand, the authors seem to be reaching further than the data suggest to explain small differences in concentrations with a complicated chemical mechanism that was vaguely speculated around Line 100 in the introduction. It's okay if the authors can't explain where these chemicals come from. The measurements are fairly novel and interesting in and of themselves. A much more detailed chemical analysis is warranted to articulate chemical mechanisms responsible for HNCO production.

**Reply:** We appreciate your valuable comments and suggestions. Indeed, key precursors and their chemical formation pathways of HNCO are still unclear in urban regions. Based on the gradient measurements, we are designing box model simulations (maybe also smog chamber experiments) to further clarify the chemical formation mechanism of HNCO.

**24.** (Line 543) Did the amides experience any tubing effects?

**Reply:** We appreciate your valuable comments and suggestions. We have provided a new figure (Figure S4) in SI to discuss the tubing effects of amides. The average mixing ratios of $C_2$-$C_{10}$ amides measured with and without the 400 m tubing agreed well within 10%. As for formamide, its concentrations measured by PTR-ToF-MS through the 400 m long tube were slightly higher than those measured without the tubing. This phenomenon was likely caused by the fact that the measurements of formamide (m/z=46) made by PTR-ToF-MS during the test were influenced by fragment ions from other species. Before the tubing test, the mixture of zero air and gas standard containing 35 VOC species were passed into the tubing. As a result, the sticky species with higher than ambient concentrations left in the tubing may influence the measurements of ambient formamide concentrations. From the measurements of other amides made with and without the 400 m long tube, we believe that using long tubes have minor effects on the measurements of amides.

[Figure]

***Figure S4.*** *Assessment of the 400 m tubing in measuring amides in ambient air. (a)*
*Average concentration ratio of amides with and without long tubing. (b) Scatterplots of*
*mixing ratios of C₃ amides measured with the 400 m long tube versus those measured*
*without the long tube.*

**25.** (Line 564) Please replace all mentions of "growth rates" with more suitable
terminology like suggested above.

   **Reply:** We appreciate your valuable suggestion and we have replaced "growth rate"
with "enhancement rate" throughout the manuscript.

**26.** (Line 574) Again, I don't really understand the utility of using the CIC calculation
to critique ground level measurements. If the authors want to make the point that

chemicals can be produced at higher altitudes and so using ground level measurements for modeling may not be adequate for answering some questions that would be reasonable.

**Reply:** We appreciate your valuable comments and suggestions. The measurement of trace gases at ground level is one of the most effective approaches to monitor atmospheric chemical processes and the formation of secondary pollution. In this study, we want to highlight that the formation mechanisms of secondary pollutants in urban regions aloft may have large differences from those at the ground level. We agree with your opinion that the ground-level measurements may not be adequate for model simulations to clarify some important questions in the formation of secondary pollution. The usage of CICs could better characterize the change in the budget of the chemical species in comparison to the ground-level measurements. We apologize for the inappropriate phrasing and have already revised the sentences in the manuscript. [see P: 23; L: 578-580]

*"The formation of some chemicals can be largely enhanced at higher altitudes and so using ground-level measurements to constrain numerical models may be not adequate."*

**27.** (Conclusions) I'm confused about the authors conclusions regarding the tubing analysis. My understanding from the data is that formic acid experiences significant tubing effects from the long lines and HNCO did not. The authors didn't really provide any information on if they had to correct the data for tubing delays or best practices when measuring formic acid from really long lines. Did the authors have to correct HCOOH concentrations for the tubing effects?

**Reply:** We appreciate your valuable comments and suggestions. Our results show that PFA tubes do affect the measurements of formic acid, especially for the tubes with hundreds of meters in length. However, based on our observations (Figure 2), the impact of long tubes on formic acid measurement is less than 20% ($k = 0.84$), and does not significantly affect the conclusions about the vertical variability of formic acid. The

vertical increasing gradients of formic acid may be slightly enhanced if the tubing effects are considered. We have revised the conclusions in the manuscript. [see P: 23-24; L: 585-587]

*"The measurements of formic acid made through long tubes were slightly influenced by the memory effect of tubing walls, and the vertical increasing gradients of formic acid may be slightly enhanced if the tubing effects were considered."*

**28.** (Line 72) HNCO is an organic acid.

**Reply:** We appreciate your valuable comments. Actually, HNCO is an inorganic acid and please refer to our reply to the comment 8.

**29.** (Line 98-102) Can the authors start this hypothesized mechanism with "We hypothesize.." or something like that to indicate that this potentially important mechanism is the authors thoughts?

**Reply:** We appreciate your valuable suggestion, and this sentence has been removed from our manuscript due to the modification of the introduction.

**30.** (Line 93-105) The authors might consider removing this paragraph. There's three sentences of loose speculation in it and I think the important point could be summarized in a single sentence tacked on to the last sentence of the previous paragraph. Basically "vertical gradient measurements of HNCO can help elucidate potential formation sources and mechanisms".

**Reply:** We greatly appreciate your comments and suggestions to improve our manuscript. We agree with your opinion that these discussions are not suitable for inclusion in the introduction. We have rewritten the paragraph in the manuscript. [see P: 5-6; L: 97-104]

*"Amides are reported to be the main precursors of isocyanic acid in urban regions (Wang et al., 2020). Isocyanic acid is the oxidative degradation product of amides initiated by OH radicals, $NO_3$, radicals, and Cl atoms (Barnes et al., 2010). In addition to primary emissions from organic solvents and various industrial processes, amides*

*can be also formed through the atmospheric accretion reactions of organic acids with amines or ammonia (Barnes et al., 2010; Yao et al., 2016). Vertical gradient measurements of HNCO can help elucidate potential formation sources and mechanisms."*

**31.** (Line 106) Replace "mass spectrometer" with "mass spectrometry"

**Reply:** Thank you for pointing out this mistake and we have revised it in the manuscript. [see P: 6; L: 105-107]

*"Chemical ionization mass spectrometry (CIMS) can effectively detect and quantify atmospheric formic and isocyanic acids (Bannan et al., 2014; Chandra and Sinha, 2016; Liggio et al., 2017; Mungall et al., 2018; Fulgham et al., 2019)."*

**32.** (Line 108-110) I would recommend removing this statement. In the following sentences the authors cite studies that provide evidence to the contrary. In 2013 a group from Cal-Tech put their CIMS instrument on top of a tower for measurements.

**Reply:** We appreciate your valuable suggestions and we have removed this sentence in the revised manuscript.

**33.** (Line 125) Remove "alkane" from PFA definition.

**Reply:** Thank you pointing out this mistake and it has been corrected in the manuscript. [see P: 6; L: 122-123]

*"In this study, we first assessed the effects of long perfluoroalkoxy (PFA) Teflon tubes on measurements of formic and isocyanic acids."*

**34.** (Line 172) Replace "ionic" with "ion molecule reaction"

**Reply:** Thank you for pointing out this mistake and it has been revised in the manuscript. [see P: 8; L: 173-174]

*"The ion molecular reaction (IMR) chamber is adjacent to the FIGAERO and utilizes a vacuum ultraviolet ion source (VUV-IS)."*

**35.** (Line 175) "MIR" should be "IMR"

**Reply:** Thank you for pointing out this mistake and we have corrected it in the manuscript. [see P: 8; L: 174-175]

*"Iodide anion (I⁻) is produced from the photoionization of methyl iodide (CH₃I) in IMR (Ji et al., 2020)."*

36. (Line 179-181) Move this sentence to Line 170 in front of "A filter inlet for gases and aerosols…"

**Reply:** We appreciate your valuable suggestion and they have been revised in the manuscript. [see P: 8; L: 168-173]

*"Due to the high sensitivity to oxygenated volatile organic compounds (OVOCs), the iodine ion source has been widely used in previous studies (Yuan et al., 2015; Schobesberger et al., 2016; Mungall et al., 2018). A Filter Inlet for Gases and AEROsols (FIGAERO) was used to perform the switch between the gas and particle measurement modes (Lopez-Hilfiker et al., 2014).*

37. (Line 201) Replace "eliminated" with "corrected for"

**Reply:** We appreciate your suggestion and they have been revised in the manuscript. [see P: 9; L: 201-203]

*"Impacts of the changes in ambient humidity on measurements of the ToF-CIMS for both formic and isocyanic acids were determined in our laboratory and were corrected for when calculating their respective concentrations."*

38. (Line 366-369) Consider rewording…

"Photochemical formation of secondary pollutants was largely suppressed from July 13th to 30th due to weak solar radiation (characterized by small j(NO2) values) and precipitation."

**Reply:** We appreciate your valuable suggestion and we have rephrased this sentence in the manuscript. [see P: 15; L: 367-369]

*"The photochemical formation of secondary pollutants was weak from July 13ᵗʰ to*

*30th due to the cloudy and rainy weather.”*

**39.** (Figure 8) Instead of putting X=1 in the figure can the authors just write in the caption that a dotted line is shown for the normalized "growth rate" of 1?

**Reply:** We appreciate your valuable suggestion. We have revised figure 8 and 9 in the manuscript and rephrased the caption.

[Figure]

*Figure 8. Normalized vertical profiles of the enhancement rate of ozone, formic acid, and isocyanic acid between 6:00-10:00 LT averaged over the whole campaign. Enhancement rate of the species at different altitudes were normalized to those at 5 m. The dotted line indicates the normalized enhancement rate of 1.*

[Figure]

***Figure 9.*** *Normalized vertical profiles of formic acid, isoprene, formaldehyde, MVK and MACR in daytime (11:00-16:00 LT) averaged over the whole campaign. Mixing ratios of the species at different altitudes were normalized to those at 5 m. The dotted line indicates the normalized concentration of 1.*

**40.** (Line 436) Consider rephrasing to "Our results point to the likely importance of photochemistry as a source of formic acid that is enhanced with increasing height within the boundary layer."

**Reply:** We appreciate your valuable suggestion and we have rephrased this sentence in the manuscript. [see P: 18; L: 436-438]

*"Our results point to the likely importance of photochemistry as a source of formic acid that is enhanced with increasing height within the boundary layer."*

**41.** (Line 445) Consider citing Link, et al. (2021).2

**Reply:** We appreciate your valuable suggestion. We have meticulously read the recommended article and cited it in the revised manuscript. [see P: 18; L: 444-446]

*"In addition, the photooxidation of other biogenic and anthropogenic hydrocarbons is also a key source of formic acid (Paulot et al., 2011; Millet et al., 2015; Link et al., 2021)"*

**42.** (Line 470) I think the authors mean Figure 5f.

**Reply:** Thank you for pointing out this mistake and we have corrected it in the manuscript. [see P: 19; L: 472-473]

*"As shown in Figure 10, the CICs of formic acid had a flatter diurnal pattern in comparison to those at ground level."*

**43.** (Figure 12) Can the authors provide a description in the caption of what the patterning indicates for the bars?

**Reply:** We appreciate your valuable suggestion. The patterns of the bars in Figure 13(b) are used to clearly distinguish the average concentrations of the species between the two heights. We have provided a description in the caption of Figure 13 to specify the patterns of the bars.

[Figure]

*Figure 13. (a) Vertical variations in composition and concentrations of amides. (b)*

*Concentration comparison of formic acid, isocyanic acid, and amides between 5 and 320 m. The data in both (a) and (b) was the average results of the whole campaign. The patterns of the bars are used to distinguish the average concentration of the species at the two heights.*

**44.** (Barnes, et al. 2010 Reference) Should the journal abbreviation be ChemPhyChem?

**Reply:** Thank you for pointing out this mistake and we have corrected it in the manuscript.

---

## Author Comment (AC2)

* * *
**Response to Reviewer #2**
* * *
**1.** This paper reported the vertical measurement of formic and isocyanic acids in a typical urban region. In this paper, the authors fully assessed the uncertainty caused by the ~400 m long tube sampling techniques, they showed the tubing has a negligible and minor influence on the formic and isocyanic acids, and confirmed the feasibility in vertical measurement by using the long PFA tube. By analyzing the vertical measurement dataset, they characterized the variation and formation of the two species on a diurnal and vertical scale and highlighted the limitations of using the ground measurement dataset to look insight to their chemical behaviors on the boundary layer scales. This paper contributed very useful sampling technique and a comprehensive monitoring dataset to atmospheric chemistry from the perspective of vertical scale, I would like to recommend this paper be published in ACP as a Measurement report type subject to the following minor comments.

**Reply:** We appreciate your valuable comments and suggestions, which are very helpful for the improvement of our manuscript.

**2.** The characterization of the tubing section is interesting and very useful to the community; thus I would like to encourage the authors to add one sentence to introduce this part in the abstract.

**Reply:** We greatly appreciate your suggestion and we have provided a brief description about the tubing assessment in the abstract. [see P: 2; L: 17-22]

*"To address this issue, we assessed the impact of long tubes on the measurement uncertainties of formic and isocyanic acids and found that the tubing impact was negligible. Then, we conducted continuous (27 days) vertical gradient measurements (five heights between 5-320 m) of formic and isocyanic acids using long tubes based on a tall tower in Beijing, China, in the summer of 2021."*

**3.** Line 27-28 and Line 29-31, is it possible to switch the order of the two sentences?

Since the daytime vertical gradient and the causes can be illustrated clearly first. By the way, the sentence in Lines 27-28 cannot be well understood as, to why the positive gradient at nighttime can be indicative of the dominating role of photochemistry in the formation of the two species.

**Reply:** We appreciate your valuable suggestions and we have rearranged the order of two sentences. During the field campaign, formic acid exhibited a significant positive gradient both during the day and at night, which indicates the enhancement of their contributions from the sources aloft. As we discussed in lines 397-399, large amounts of these two species will accumulate near the surface with strong negative vertical gradients if primary emissions dominate its contributions, as manifested by vertical toluene profiles. [see P: 2; L: 29-34]

*"The positive vertical gradients of formic and isocyanic acids in daytime imply the enhancement of their secondary formation in urban regions aloft, predominantly due to the enhancements of oxygenated volatile organic compounds. Furthermore, the afternoon peaks and positive vertical gradients of formic and isocyanic acids in nighttime also indicate their minor contributions from primary emissions from ground-level sources."*

**4.** Figure 3b, what happened in the result of 200 m with a much higher k and R2?

**Reply:** Thank you for your valuable comments. During the test of the 200 m tubing, meteorological conditions significantly changed with lower temperatures and stronger winds in comparison to the days on which the tests of the other lengths of tubes were performed. As shown in Figure A1, the concentrations of formic acid and isocyanic acid were evidently enhanced and significantly varied during the 400 m tubing test. In contrast, ambient concentrations of formic and isocyanic acid were relatively low and slightly varied, resulting in the exceedingly large or low values of k and $R^2$ between the concentrations of formic acid measured with and without the 200 m long tubing. However, according to the results of the test, the average concentration difference of formic and isocyanic acid measured with and without the 200 m tubing agreed well

within 4%, suggesting that the 200 m long tube has minor effects on the measurements of formic and isocyanic acid.

[Figure]

***Figure A1.*** *Time series of (a) formic and (b) isocyanic acid concentrations measured with and without the 400 m long tube.*

[Figure]

*Figure A2. Time series of (a) formic and (b) isocyanic acid concentrations measured with and without the 200 m long tube.*

**5.** According to Figure 2, I propose that the impact of formic and isocyanic acids is quite similar, the difference may be caused by the ambient concentration of the two species, even if the conclusion is not changed that the influence is small.

**Reply:** We appreciate your valuable comments. We agree with your opinion that the effects of long tubes on measurements of both formic and isocyanic acids were similarly affected by the changes in their ambient concentrations. As we discussed in the manuscript, formic acid exhibited more significant variation in ambient concentration than isocyanic acid, and therefore, it was more affected by the usage of long tubes.

---

## Author Response (AR2)

* * *
**Response to Editor**
* * *
**1.** Thank you for your consideration of the referee comments. Many of the comments have been addressed. However, please address the following comments prior to publication.

**Reply:** We appreciate your valuable comments and suggestions, which are very helpful for the further improvement of our manuscript.

**2.** Sect 2.2: Please indicate if the sampling tubes were covered (to prevent photochemistry of deposited species from causing interferences) and implications if not.

**Reply:** We appreciate your valuable suggestions. During the field campaign, the sampling tubes were not covered by opaque materials but were installed inside the iron tower to avoid direct sunlight. We have provided these details in the revised manuscript. [see P: 7; L: 152-153]

*"All the sampling tubes were installed inside the iron tower to avoid direct sunlight."*

**3.** Lines 217-219: Indicate where the zero air was added to perform the blank. Was the blank just of the instrument or of the instrument + the sampling lines?

**Reply:** We appreciate your valuable comments and suggestions. During the field campaign, blank measurements of the instrument were made by adding zero air to the inlet of the instrument without through the long tubes. We have conducted tests to investigate impacts of the long tubes on the blank measurements of the instrument. Background signals of the instrument made through the 400 m long tube for formic acid and isocyanic acid at a zero air flow rate of 13 SLPM have been provided in Figure S3 in SI. The difference between the average signals of formic acid measured with and without the 400 m long tube was only 5 ncps, accounting for a fraction of 1.4% in the sensitivity of formic acid (357.1 ncps/ppbv) and thus can be ignored. The average difference of the instrument background signal for isocyanic acid measured with and

without the 400 m long tube was only 0.03 ncps, accounting for a very tiny fraction of 0.05% in the sensitivity of isocyanic acid (51.4 ncps/ppbv) and thus can also be ignored. These results indicate that the usage of the long tubes have minor effects on the blank measurements of the instrument for both formic acid and isocyanic acid. We have provided an additional discussion in the revised manuscript to tell the readers that the blank measurements of the instrument were made without through the long tubes. [see P: 9; L: 185-189]

*"As shown in Figure S3, there was no significant difference between the background signals of the instrument made with and without the long tubes. Therefore, blank measurements of the instrument were made by adding zero air just to the inlet of the instrument without through the long tubes during the field campaign."*

A detailed description of the test results is provided in SI. [see P: 3; L: 22-30]

*"Figure S3 shows the background signals of the instrument made through the 400 m long tube for formic acid and isocyanic acid at a zero air flow rate of 13 SLPM. The difference between the average signals of formic acid with and without the 400 m long tube was only 5 ncps, accounting for a fraction of 1.4% in the sensitivity of formic acid (357.1 ncps/ppbv) and thus can be ignored. The average signal difference for isocyanic acid with and without 400 m long tube was only 0.03 ncps, accounting for a very tiny fraction of 0.05% in the sensitivity of isocyanic acid (51.4 ncps/ppbv) and thus can also be ignored. These results indicated that the usage of the long tubes had minor effects on the blank measurements of the instrument for both formic acid and isocyanic acid."*

[Figure]

***Figure S3.*** *Time series of (a) formic and (b) isocyanic acid blank signals measured with and without the 400 m long tube at a zero air flow rate of 13 SLPM.*

**4.** Referee 1 comment 10: Please expand on the discussion about how the delay time can be shorter than the residence time and incorporate this into the manuscript. While it may be discussed in a previous manuscript, it is central to this paper as well and should be included in at least a brief format.

**Reply:** We appreciate your valuable comments and suggestions. Residence time refers to the time required for the sample gas to pass through the tubes. As for the measured tubing delays of trace gases, they refer to the amounts of time required for the instruments to measure stable concentrations of targeted species in response to a change in species concentrations at the tubing inlet. Residence time is the same for all trace gases, depending on the length of the tube, the diameter of the tube, and the flow rate of the sample gas. However, the tubing delay of trace gases are different and depend on the flow rate, their respective saturated concentrations/Henry's constants, and molecular diffusion rates. We have provided related definitions and discussions in the revised manuscript. [see P: 13; L: 297-307]

*"The delay time of formic acid mentioned here is different from the residence time of the gas through the long tubing. Residence time refers to the time required for the sample gas to pass through the tubes. As for the measured tubing delays of trace gases, they refer to the amounts of time required for the instruments to measure stable concentrations of targeted species in response to a change in species concentrations at the tubing inlet. The residence time is the same for all trace gases, depending on the length of the long tube, the inner diameter of the tube, and the flow rate of the sample gas. However, the tubing delay for each trace gas is different and depends on the flow rate, their respective saturated concentrations/Henry's constants, and molecular diffusion and diffusion rates. The difference between residence time and delay time is also discussed in detail in our previous work (Li et al., 2023)."*

*References:*

*(1) Li, X., Zhang, C., Liu, A., Yuan, B., Yang, H., Liu, C., Wang, S., Huangfu, Y., Qi, J., Liu, Z., He, X., Song, X., Chen, Y., Peng, Y., Zhang, X., Zheng, E., Yang, L., Yang, Q., Qin, G., Zhou, J., and Shao, M.: Emerging investigator series: Assessment of Long Tubing in Measuring Atmospheric Trace Gases: Applications on Tall Towers, Environmental Science: Atmospheres, 10.1039/d2ea00110a, 2023.*

**5.** Line 287: What mixing ratios were used for the step-function change? Please include that in the text.

**Reply:** We appreciate your valuable comments and suggestions. The step-function change used in our study is 7.5-0 ppbv. During the test, a mixture of zero air and formic acid vapor with a formic acid concentration of 7.5 ppbv was first introduced into the 400 m long tube. After the stabilization of the measured formic acid signal, we stopped the supply of the formic acid vapor and only zero air was introduced into the instrument at the same flow rate. Then, the measured signals of formic acid through the long tube declined and we can obtain a depassivation curve of formic acid to calculate the tubing delay. We have rephrased this sentence in the revised manuscript to make it clearer. [see P: 11; L: 253-256]

*"The depassivation curve of formic acid measured at the outlet end of the long tubing was used to calculate its tubing delay and was obtained by using a step-function change of the formic acid concentration from 7.5 ppbv to 0 ppbv at the tubing inlet (Pagonis et al., 2017; Deming et al., 2019)."*

**6.** Line 352: Why is delta t defined here – it isn't used in equation 1.

**Reply:** Thank you for pointing out this mistake and we have adjusted the position of this sentence in the manuscript. [see P: 14; L: 333-335]

*"*

$$\Delta[X]_t = [X_{with}]_t - \frac{\sum_{t-\Delta t}^{t}[X_{with}]}{\Delta t} \tag{2}$$

*where $\Delta t$ is the change in time relative to time t and was used to characterize the influential time of the memory effect."*

**7.** Lines 386-388: I think this last sentence over simplifies the formic acid discussion and may cause confusion to readers. I recommend clarifying that formic acid does suffer from memory effects that need to be considered when interpreting the measurements.

**Reply:** We appreciate your valuable comments and suggestions. The test results confirmed that the measurements of formic acid and isocyanic acid through long tubes can be used to characterize their vertical and temporal variability. However, a further correction of the formic acid measurements made through the long tubes must be performed if they were used to accurately calculate the kinetic parameters of chemical reactions regarding the formation and removal of formic acid at different heights. We have rewritten this sentence in the manuscript to make it clearer. [see P: 15-16; L: 375-380]

*"The test results confirmed that the measurements of formic acid and isocyanic acid through long tubes can be used to characterize their vertical and temporal variability. However, a further correction of the formic acid measurements made through the long tubes must be performed if they were used to accurately calculate the kinetic parameters of chemical reactions regarding the formation and removal of*

*formic acid at different heights."*

**8.** Referee 1 comment 14/Referee 2 comment 4: Please include the information about the 200 m tubing test in the manuscript to explain why the 200 m appears to be an outlier in Fig. 3.

   **Reply:** We appreciate your valuable comments and suggestions. During the test of the 200 m tubing, meteorological conditions significantly changed with lower temperatures and stronger winds in comparison to the days on which the tests of the other lengths of tubes were performed. As shown in Figure S5, the concentrations of formic acid and isocyanic acid were evidently enhanced and significantly varied during the 400 m tubing test. In contrast, ambient concentrations of formic and isocyanic acid were relatively low and slightly varied, resulting in the exceedingly large or low values of $k$ and $R^2$ between the concentrations of formic acid measured with and without the 200 m long tubing. However, according to the results of the test, the average concentration difference of formic and isocyanic acid measured with and without the 200 m tubing agreed well within 4%, suggesting that the 200 m long tube has minor effects on the measurements of formic and isocyanic acid. We have provided these results and discussions in the revised manuscript. [see P: 15; L: 351-362]

   *"During the test of the 200 m tubing, meteorological conditions significantly changed with lower temperatures and stronger winds in comparison to the days on which the tests of the other lengths of tubes were performed. As shown in Figure S5, the concentrations of formic acid and isocyanic acid were evidently enhanced and significantly varied during the 400 m tubing test. In contrast, ambient concentrations of formic and isocyanic acid were relatively low and slightly varied, resulting in the exceedingly large or low values of k and R² between the concentrations of formic acid measured with and without the 200 m long tubing. However, according to the results of the test, the average concentration difference of formic and isocyanic acid measured with and without the 200 m tubing agreed well within 4%, suggesting that the 200 m long tube has minor effects on the measurements of formic and isocyanic acid."*

[Figure]

***Figure S5.*** *Time series of (a) formic and (b) isocyanic acid concentrations measured with and without the 200 m long tube.*

**9.** How was the PBL height measured?

**Reply:** We appreciate your valuable comment. The planetary boundary layer height (PBLH) is the reanalysis data obtained from the website of the NOAA Air Resources Laboratory (https://ready.arl.noaa.gov/READYamet.php). We have provided an additional description on the source of the PBLH data in the revised manuscript. [see P: 10; L: 220-224]

*"The planetary boundary layer height (PBLH) data was obtained from the web portal of the Real-time Environmental Applications and Display sYstem (READY) of the National Oceanic and Atmospheric Administration (NOAA) Air Resource Laboratory (https://ready.arl.noaa.gov/READYamet.php)."*

**10.** Lines 521-524: The CICs consider the residual layer which will not have depositional loss occurring. The ground-level measurements will have stronger depositional loss at night. I'm not sure how looking at ground-level measurements is

thus overestimating removal. In fact, it seems like considering CICs will underestimate the removal. Please clarify.

**Reply:** We appreciate your valuable comments and agree with your opinion that ground-level measurements are more affected by depositional losses, while such losses in the residual layer are minor. By using CICs, we want to highlight the change in the total budget of formic acid within the entire boundary layer. The residual layer is formed due to the shrimp of the boundary layer and thus can be considered part of the boundary layer. Due to the absence of depositional loss, large amounts of formic acid retained in the residual layer during the night can entrained into the boundary layerduring the daytime. If the removal rates of formic acid from ground-level measurements are used to reflect those at high altitudes (e.g., in the residual layer), the removal of formic acid in the entire atmospheric boundary layer will be overestimated. We have rewritten this sentence in the manuscript to make it clearer. [see P: 20; L: 508-514]

*"The ground-level measurements were more affected by depositional losses, while such depositional losses in the residual layer were nearly absent. However, the chemical species retained in the residual layer were closely related to their budgets in the daytime boundary layer. If the removal rates of formic acid from ground-level measurements were used to characterize those at high altitudes (e.g., in the residual layer), the removal of formic acid in the entire boundary layer would be overestimated."*